# Integrative Approaches in Acute Ischemic Stroke: From Symptom Recognition to Future Innovations

**DOI:** 10.3390/biomedicines11102617

**Published:** 2023-09-23

**Authors:** Vicentiu Mircea Saceleanu, Corneliu Toader, Horia Ples, Razvan-Adrian Covache-Busuioc, Horia Petre Costin, Bogdan-Gabriel Bratu, David-Ioan Dumitrascu, Andrei Bordeianu, Antonio Daniel Corlatescu, Alexandru Vlad Ciurea

**Affiliations:** 1Neurosurgery Department, Sibiu County Emergency Hospital, 550245 Sibiu, Romania; vicentiu.saceleanu@gmail.com; 2Neurosurgery Department, “Lucian Blaga” University of Medicine, 550024 Sibiu, Romania; 3Department of Neurosurgery, “Carol Davila” University of Medicine and Pharmacy, 020021 Bucharest, Romania; razvan-adrian.covache-busuioc0720@stud.umfcd.ro (R.-A.C.-B.); horia-petre.costin0720@stud.umfcd.ro (H.P.C.); bogdan.bratu@stud.umfcd.ro (B.-G.B.); david-ioan.dumitrascu0720@stud.umfcd.ro (D.-I.D.); andrei.bordeianu@stud.umfcd.ro (A.B.); antonio.corlatescu0920@stud.umfcd.ro (A.D.C.); prof.avciurea@gmail.com (A.V.C.); 4Department of Vascular Neurosurgery, National Institute of Neurology and Neurovascular Diseases, 020022 Bucharest, Romania; 5Centre for Cognitive Research in Neuropsychiatric Pathology (NeuroPsy-Cog), “Victor Babes” University of Medicine and Pharmacy, 300736 Timisoara, Romania; 6Department of Neurosurgery, “Victor Babes” University of Medicine and Pharmacy, 300041 Timisoara, Romania; 7Neurosurgery Department, Sanador Clinical Hospital, 010991 Bucharest, Romania

**Keywords:** acute ischemic stroke, cerebrovascular disease, stroke onset symptoms, neuroimaging, neuroradiology, endovascular thrombectomy, neurovascular treatment, revascularization, fibrinolytic therapy

## Abstract

Among the high prevalence of cerebrovascular diseases nowadays, acute ischemic stroke stands out, representing a significant worldwide health issue with important socio-economic implications. Prompt diagnosis and intervention are important milestones for the management of this multifaceted pathology, making understanding the various stroke-onset symptoms crucial. A key role in acute ischemic stroke management is emphasizing the essential role of a multi-disciplinary team, therefore, increasing the efficiency of recognition and treatment. Neuroimaging and neuroradiology have evolved dramatically over the years, with multiple approaches that provide a higher understanding of the morphological aspects as well as timely recognition of cerebral artery occlusions for effective therapy planning. Regarding the treatment matter, the pharmacological approach, particularly fibrinolytic therapy, has its merits and challenges. Endovascular thrombectomy, a game-changer in stroke management, has witnessed significant advances, with technologies like stent retrievers and aspiration catheters playing pivotal roles. For select patients, combining pharmacological and endovascular strategies offers evidence-backed benefits. The aim of our comprehensive study on acute ischemic stroke is to efficiently compare the current therapies, recognize novel possibilities from the literature, and describe the state of the art in the interdisciplinary approach to acute ischemic stroke. As we aspire for holistic patient management, the emphasis is not just on medical intervention but also on physical therapy, mental health, and community engagement. The future holds promising innovations, with artificial intelligence poised to reshape stroke diagnostics and treatments. Bridging the gap between groundbreaking research and clinical practice remains a challenge, urging continuous collaboration and research.

## 1. Introduction

Cerebrovascular diseases encompass a range of conditions resulting from pathological changes in the cerebral blood vessels that lead to brain dysfunction. These dysfunctions can arise from issues such as vascular occlusion, stenosis, rupture, malformation, wall damage, or permeability alterations. Globally, these diseases are a major contributor to mortality and disability. In fact, the economic impact of cerebrovascular disease surpasses that of many other vascular conditions. Without timely preventive actions, healthcare costs for addressing these diseases could see a threefold increase [1]. Stroke, as the primary clinical manifestation of cerebrovascular disease, poses a significant strain on global healthcare resources [2]. Data from *The Global Burden of Disease Study* reveals that stroke is responsible for 11.6% of worldwide deaths, ranking as the second leading cause of death and the third primary cause of disability [3]. Despite a notable 25% reduction in global stroke deaths recently, there was an alarming rise in the total number of stroke cases, their prevalence, mortality, and disability-adjusted life years between 1990 and 2019 [4].

Stroke represents the most frequent outcome of cerebrovascular disease and arises when an artery feeding the brain becomes blocked or ruptures. This chapter initially delves into the influence of epidemiological research on our understanding of stroke. It encompasses insights into its clinical presentation, diagnosis, and treatment, as well as the associated burden, risk factors, and repercussions. We will also explore transient ischemic attack (TIA), a condition as dangerous as stroke, and its impact on stroke risk [5].

While strokes are widely recognized, cerebrovascular diseases include a broader spectrum of conditions where the brain’s blood vessels are the primary focus [6]. Examples include subarachnoid hemorrhages, arterial dissections, CADASIL, arteriovenous malformations, venous sinus thrombosis, moyamoya disease, and vasculitis. This chapter’s latter section will delve into these diseases, emphasizing their epidemiological aspects [7].

Not all cerebrovascular anomalies produce evident symptoms immediately. Conditions such as silent brain infarcts, white-matter lesions, and microbleeds exemplify this subclinical cerebrovascular disease category [8]. Epidemiological imaging studies aim to gauge the prevalence of these subclinical conditions in the general populace, along with their risk factors and consequences [7].

Globally, stroke stands as the second primary cause of both disability and death. Low- and middle-income countries bear the brunt of this affliction. In 2016 alone, the world saw 13.7 million new stroke cases, with approximately 87% being ischemic strokes. Within these, by a modest estimate, between 10% and 20% were due to LVO. When considering treatment, less than 5% of acute ischemic stroke patients globally underwent IVT within the appropriate therapeutic time frame, and under 100,000 MTs were carried out in that year. These figures underscore the significant disparity between the number of eligible patients and the relatively scant utilization of these advanced treatments worldwide. However, several global efforts are in motion to research interventions that could potentially enhance the care systems and bridge this disparity [9,10].

From the 1980s onward, the occurrence of ischemic stroke among young adults has been on an upward trend. This uptick correlates with a rise in vascular risk factors and substance abuse in younger individuals. Unlike their older counterparts, young adults have a broader spectrum of risk factors, which include age-specific elements like pregnancy/puerperium and the use of oral contraceptives. Lifestyle choices, such as sedentary behavior, excessive alcohol intake, and smoking, also play a role [11]. In addition to these, over 150 distinct causes of early-onset ischemic stroke are recognized, inclusive of rare monogenic disorders. Recently, there have been strides in the diagnosis and treatment of stroke in young adults. This includes the molecular identification of monogenic vasculitis due to an adenosine deaminase 2 deficiency and the transcatheter closure of the patent foramen ovale for secondary prevention. Compared to their peers of similar age and gender, these patients face a mortality rate that is four times higher, predominantly due to cardiovascular causes. Furthermore, up to 15% of these patients risk a recurrent stroke within a decade [12] (pp. 2007–2018). Specific subgroups, especially those with atherosclerosis, high-risk cardioembolic sources, and small vessel disease, are more vulnerable to adverse outcomes such as survival and recurrent vascular incidents. These young stroke survivors are also susceptible to other long-term complications like epilepsy, pain, cognitive challenges, and depression [13].

Globally, stroke ranks as the second and third foremost cause of death and disability, respectively. In the global context, 68% of all strokes are ischemic, and 32% are hemorrhagic. However, in the USA, the statistics slightly vary, with 87% of strokes being ischemic, 10% hemorrhagic, and roughly 3% categorized as subarachnoid hemorrhage [14]. While comprehensive stroke data for India remains limited, western data offers some extrapolation points. Banerjee et al.’s 2001 study highlighted that India had a crude stroke prevalence rate of 147/100,000 and an annual incidence rate of 36/100,000. Notably, women exhibited higher age-adjusted prevalence (564/100,000 for women versus 196/100,000 for men) and incidence rates (204/100,000 for women compared to 36/100,000 for men). In diverse studies, the overall stroke prevalence in India fluctuated between 147–922/100,000 [15].

On a global level, stroke has far-reaching socio-economic and health implications.

In the realm of clinical practice, susceptibility-weighted imaging (SWI) emerges as an invaluable imaging method, particularly in identifying intracerebral hemorrhage, intra-arterial thrombus, microbleed, and the hemorrhagic transformation of acute stroke. Prior research reveals that the conspicuous cortical vessels detected via SWI (PCV–SWI)—which pinpoint the cortical pia mater—mirror the scope of hypoperfusion and share a relationship with leptomeningeal collaterals in AIS. There have been efforts to ascertain the merit of PCV–SWI in gauging the state of collaterals and projecting outcomes in AIS. However, the broader clinical utility of PCV–SWI for assessing leptomeningeal collaterals prior to employing recanalization therapy is yet to reach consensus. These investigations often differ in design, patient traits, imaging techniques for estimating collateral, and often involve limited sample sizes [16]. A pivotal query remains: Can PCV–SWI serve as an alternative to mCTA for evaluating leptomeningeal collaterals and predicting patient outcomes post-recanalization therapy? [17].

Regarding the Podlaskie Voivodeship region in Poland, comprehensive examinations analyzing toxic metals in AIS patients appear scant. Our current findings underline the critical public health ramifications, spotlighting a notable trace element imbalance in acute ischemic stroke patients [18]. Of notable significance from our study was the discovery that even at moderately low exposure levels, elevated blood Cd concentrations substantially influence ischemic stroke onset. Noteworthy differences in molar ratios between AIS patients and controls were also observed, with smokers demonstrating a higher propensity for LAA etiology. Despite this, Pb blood concentrations did not exhibit significant variations between the two groups. The growing nexus between environmental contaminants and stroke has piqued medical interest [19]. Research is increasingly showcasing heavy metal exposure as a potential stroke risk factor, with emerging studies linking elevated blood-Cd levels to higher stroke occurrences. A recent systematic review and meta-analysis even suggest chronic exposure to metals like Pb, Cd, and Cu might elevate stroke risk [18].

Cerebral collaterals undeniably influence the sustenance of penumbral tissue following an acute ischemic stroke. While recent research accentuates the role of collaterals in determining whether acute ischemic stroke patients are fit for reperfusion therapy, a comprehensive grasp on their predictive importance remains elusive [20]. Our literature review concentrated on methodologies for collateral assessment and their significance in acute ischemic stroke patients undergoing reperfusion therapy. Both the therapeutic and prognostic values of collaterals in acute ischemic stroke, in the context of both intravenous thrombolysis and endovascular therapy, are encapsulated. Prospects for future studies and potential drug interventions targeting collateral enhancement are also explored. Collaterals could crucially help identify acute ischemic stroke patients poised to gain from endovascular treatment in an extended timeframe. A challenging situation in acute ischemic stroke is the collateral circles, which are not determined by clinical factors or demographic influences but are highly associated with preliminary cerebrovascular events [21]. A renewed focus on understanding the role of collaterals in acute ischemia is essential, with clinical research needed to demarcate its significance in patient selection and prognosis for acute stroke [22].

A multitude of factors play into the issue of prehospital delay. Among them, socioeconomic status (SES) has been identified by some researchers as a potential catalyst for this delay [23,24]. A handful of studies have illuminated a palpable link between individual SES and prehospital delay [25,26,27]. The relationship between community- or neighborhood-level SES and the same delay in AIS patients has also been probed, but the findings have not always aligned [28,29]. One plausible explanation for these disparities may lie in urbanization. Urbanization is more than just the physical migration to cities; it encompasses shifts in economic, social, and cultural facets of society. Given the variance in SES distribution across urban and rural zones, a nuanced, stratified analysis anchored on urbanization status is essential. When the broader picture is considered, area-level SES does not seem to correlate with prehospital delay in AIS patients post-covariate adjustments. However, living in socioeconomically deprived urban settings could potentially amplify the delay—a pattern not mirrored in rural locales. Urban SES, therefore, might be a roadblock to curbing prehospital delay in AIS patients. Tying it all together, the utilization of emergency medical services (EMS) stands out as a vital component in mitigating prehospital delays, irrespective of urbanization or SES gradients. For a more comprehensive approach, EMS staff should be vigilant about potential biases or stereotypes they might harbor towards low-SES patients [30].

Diving deeper into the topic, the pressing need for timely diagnoses and swift interventions cannot be overstated. This meta-analysis uncovered a substantial link between stroke etiology and the status of cerebral collaterals prior to intervention in AIS patients set for reperfusion therapy (RT). It was observed that large artery atherosclerosis (LAA) correlated with enhanced rates of beneficial pre-intervention collateral, while cardioembolic (CE) strokes were associated more with deficient pre-intervention collateral. The collateral status of AIS stands as a significant determinant in influencing post-RT outcomes [22,31]. While past meta-analyses endeavored to spotlight collateral status as an outcome predictor for endovascular stroke treatment, this particular analysis is pioneering in its effort to juxtapose collateral status with stroke etiology [32,33].

Delving into the genesis of cerebral collaterals, several environmental factors come into play, with the presence of atherosclerotic blockages that hamper cerebral blood flow standing out. Such obstructions tweak vessel dynamics, escalating shear pressure and activating specific cellular pathways that pave the way for collateral creation and vascular restructuring [34]. This mechanism aligns with findings by Rebello et al., which showcased AIS patients plagued by cervical atherosclerotic issues as having a favorable pre-intervention collateral status in contrast to those afflicted by embolic strokes due to atrial fibrillation [35]. Hassler et al. further bolster this theory by pinpointing how a pre-existing atherosclerotic blockage in the carotid artery positively correlates with improved collateral health [36]. This observation resonates with this meta-analysis, where LAA was distinctly linked with pre-intervention collateral health in AIS patients [37].

Zeroing in on patient treatment, it was noted that individuals subjected to dual causal treatments showcased more severe stroke symptoms than their counterparts, as indicated by the elevated national institutes of health stroke scale (NIHSS) scores both at the time of admission and 24 h post-diagnosis. These patients underwent thrombolytic treatments, complemented by an imperative thrombectomy. Presently, stent retrievers dominate as the cornerstone tool for thrombectomies, engineered for optimum adhesion and penetration within thromboembolic matter in arteries [38]. But it is important to note that thrombectomy, despite being a mechanical procedure, has cellular repercussions. This might play into triggering an augmented inflammatory response, evident in heightened neutrophil counts, resulting in an elevated neutrophil-to-lymphocyte ratio (NLR) among these patients [39].

In the realm of understanding the pathological intricacies of ischemic strokes, the process of diagnosis predominantly hinges on imaging techniques, primarily the use of head computer tomography followed by angiography [40]. The aftermath of an acute ischemic stroke, as current knowledge indicates, can profoundly alter multiple facets of a patient’s health and overall well-being. Hence, a multidisciplinary approach is vital for evaluating the effectiveness of treatments and overseeing patient recovery. Interestingly, while advanced imaging tools offer a robust diagnosis mechanism, they often falter when detecting early-stage or mild ischemic strokes. Given the time-sensitive nature of stroke management—where every passing moment can be decisive for the patient—there is an evident demand for swift, non-invasive molecular tests that can accurately pinpoint the occurrence of an ischemic stroke. Over the years, an array of biochemical biomarkers has emerged as potential diagnostic tools. Nevertheless, many of these biomarkers have been sidelined due to challenges such as specificity concerns or fluctuations in protein biomarker dynamics, particularly post-thrombolytic therapy interventions [41,42]. Still, considering the multifaceted nature of acute ischemic stroke pathophysiology—which encompasses elements like thromboembolism, inflammation, oxidative stress, and metabolic alterations—it is plausible that pertinent molecular biomarkers could be gleaned from these varied components [43,44,45].

Ischemic strokes predominantly arise from events such as atherogenesis or thrombogenesis, which precipitate arterial blockages. These obstructions can drastically curtail or entirely halt the blood flow to specific brain regions, culminating in tissue degradation and cellular death. While the precise role of oxidative stress in instigating the ischemic process is yet to be unequivocally established, the prevailing consensus posits that an accumulation of reactive oxygen species—borne from an ischemia-induced oxidative imbalance—fuels oxidative stress, thereby driving associated cellular and molecular harm. In this context, gauging oxidative stress biomarkers could not only shed light on the underpinnings of ischemic strokes but also present diagnostic and prognostic tools with immense potential. Furthermore, these biomarkers might illuminate novel avenues for crafting antioxidant-centric therapeutic strategies [46]. Nonetheless, it is imperative to understand that, despite the promise they hold, existing diagnostic measures for ischemic strokes are not without flaws. The path to fully harnessing the diagnostic and prognostic prowess of oxidative stress biomarkers is still strewn with hurdles, primarily due to their non-specific nature. Concurrently, while a slew of research underscores the prospective merits of modulating oxidative homeostasis in ischemic stroke intervention and prevention, the dearth of comprehensive clinical studies—and the incongruous outcomes from the ones that do exist—cloud the establishment of concrete conclusions [47].

## 2. Onset and Recognition

A deep dive into the symptoms of stroke onset reveals variations influenced by the specific cerebral regions impacted. It is imperative to clarify that our position is not to advocate for pharmacotherapy to supersede the well-established speech and language therapy (SLT) in aphasia treatment [48]. Over the years, a multitude of studies have endorsed the significance of SLT, while research scrutinizing the effectiveness of pharmacotherapy has been somewhat limited in comparison [49]. The primary objective of this review is to curate a comprehensive analytical framework to discern the efficacy of cholinergic agents in aphasia. This is achieved by melding insights from neuroanatomy, neurophysiology, and neuropsychology. We are also keen to draw connections between cholinergic networks and the extensive brain regions tied to not just language but other cognitive facets that are closely aligned with linguistic abilities, such as components of working memory. Exploring the neural underpinnings of pharmacological enhancements in congruence with the neurobiology of language can potentially galvanize research on aphasia-targeted drugs. Such endeavors, if successful, could eventually revolutionize treatment avenues for individuals grappling with acquired language disorders. Investigations into both human and other primates’ brains reveal that cholinergic input extends to all cortical areas, primarily emanating from the nucleus basalis of Meynert in the basal forebrain [50]. The brain boasts eight primary cholinergic cell groups that channel projections to various regions. Each of these groups, from Ch1 to Ch8, is linked to specific nuclei. Intriguingly, the Ch1–Ch4 groups stand out as the lone neurons in the adult human central nervous system that consistently manifest significant quantities of the NGF receptor [51].

Acute stroke represents a major health challenge, not just in the United States but across the globe. The US, for instance, reports an alarming 750,000-plus stroke cases annually, leading to around 140,000 fatalities [52,53] (pp. 2000–2015). The standard protocol for patients who present with acute stroke symptoms at emergency departments in the US typically involves immediate computed tomography (CT) scanning, often prioritized over an in-depth clinical evaluation. Given its prominence in early stroke detection, imaging assumes a pivotal role. Beyond mere identification, imaging can illuminate patterns of infarction that might hint at potential causes, thereby shaping immediate interventions and refining secondary preventive measures to avert future strokes [54].

A pivotal metric in this discussion is the prehospital delay (PHD), quantified in minutes from the onset of stroke symptoms to a patient’s hospital admission. The inception of a stroke is marked by the first instance in which a neurological deficiency is discerned, either by the patient or an observer. Hospital arrival time is typically noted from the earliest timestamp documented in the emergency department’s electronic medical record. For analytical precision, PHD is bifurcated into two subintervals: the decision delay (DD) encapsulates the time elapsed between symptom onset and the initial outreach for medical intervention. The subsequent phase, or the transport delay (TD), spans the time from seeking help to the eventual hospital admission [55].

The evaluation spanned five key areas: socio-demographic attributes, clinical specifics, patients’ behavioral reactions to symptoms, cognitive reactions to the onset of symptoms, and the circumstances surrounding the stroke event. During the admission phase, the severity of the stroke was gauged using the “National Institute of Health Stroke Scale” (NIHSS). Based on this scale, strokes were categorized as either mild to moderate (a score of ≤16) or severe (a score of >16). In instances where a score was not immediately available, a retrospective examination of the patient’s documented neurological evaluations upon hospital entry was undertaken to deduce the score [56]. Prior assessments of disability employed the “modified Rankin scale”, differentiating patients as either independently functional (≤2) or reliant on assistance (>2). To discern coping mechanisms, the COPE-28 questionnaire in its situational variant was employed. Self-assessed severity perception and anxiety levels were gauged via a five-tier Likert scale. A patient’s prior awareness of strokes was gauged based on their familiarity with at least two symptomatic signs and two potential risk factors. To assess their responses, three potential answers were provided, only one of which was accurate. The onset date of the stroke was segmented into workdays (Monday through Friday) and weekends (comprising Saturdays, Sundays, and recognized holidays). Similarly, the onset time was divided into morning (06:00–14:00), afternoon (14:00–22:00), and nighttime (22:00–06:00) slots. Geographical categorizations were made using terms like ‘rural’ and ‘urban’, with the former referring to areas outside the city’s confines where the medical facility was situated and the latter describing regions within said boundary. Modes of transportation were demarcated as either ambulances or other forms [57].

Posterior circulation strokes (PCS) account for nearly 20% of the total ischemic stroke incidents [58]. However, among patients who undergo reperfusion therapy (RT), PCS figures are relatively diminutive, ranging from 5 to 19% [59]. This discrepancy stems from the unique anatomical and hemodynamic characteristics of the posterior circulation. Features such as reduced flow speeds, variance in vessel diameter, and even disparities in clot formations contribute to distinct stroke origins and progressions in comparison to their anterior circulation counterparts [60,61]. It is worth noting that the standard FAST stroke recognition tool, which comprises face asymmetry, arm weakness, and speech irregularities, does not consistently detect PCS. In fact, about 40% of PCS cases slip through this tool’s radar, whereas the BEFAST tool, which integrates balance and eye symptoms, boasts enhanced sensitivity [62,63]. PCS frequently presents with atypical symptoms like nausea and seizures, contrasting with anterior circulation strokes (ACS) [64]. This discrepancy poses challenges, as PCS sufferers often face delayed hospital admissions and diminished thrombolysis application rates. Moreover, PCS patients tend to encounter protracted management processes and experience tardier RT applications compared to ACS patients [65,66]. A noteworthy point is that the NIHSS does not encompass all PCS symptoms, occasionally leading to hesitations in applying RT to PCS. The nuanced presentations and typically lower NIHSS scores for PCS make the task of balancing RT’s effectiveness against its potential risks a complex endeavor. Although PCS’s chances of symptomatic intracerebral hemorrhaging are inferior to ACS, the perennial challenge of weighing RT risks against its benefits remains, especially when confronted with low NIHSS readings [67]. In such scenarios, decision-making often leans on nuanced, frequently subjective indicators that might not be explicitly addressed in standardized guidelines [68].

Studies have showcased diverse symptom presentations by analyzing different case studies. Within the reviewed literature concerning immersive virtual reality (IVR), no severe adverse reactions were identified. However, some reports from the broader IVR landscape have highlighted minor symptoms like dizziness, feelings of nausea, eye discomfort, and a sense of disorientation. As Tsirlin et al. elucidated in their comprehensive review [69], when designing virtual reality (VR) tools, ergonomic design is paramount. This is especially relevant when catering to post-stroke patients, who often grapple with specific challenges like restricted mobility. An intriguing observation from our review was that five studies highlighted the usage of VR training while the participants were seated—be it in a wheelchair or a regular chair [17]. Such details underscore the constraints, like limited postural stability and movement restrictions, that patients face post-stroke, suggesting the prudence of implementing IVR within supervised clinical settings. Even though the potential exists for IVR to be introduced into home settings, the prevailing evidence underscores its utility in regulated environments. Any shift towards domestic usage mandates rigorous safety evaluations [70].

When examining the literature concerning exertional heat stroke (EHS) survival, a significant portion is derived from case reports and series. Given the ethical constraints of inducing EHS in humans for study purposes, researchers predominantly rely on these case studies for insights. In our pursuit of a structured assessment, our team turned to the Joanna Briggs Institute (JBI) and its critical appraisal tools, which were tailored for case reports and series [71,72,73]. These tools, devised by the JBI, offer quality benchmarks that enhance the rigor of systematic reviews, evaluating parameters like diagnostic clarity, treatment interventions, and documented adverse events. In the context of this review, these tools proved invaluable in cherry-picking case studies and series with meticulous documentation to ascertain potential medical complications tied to EHS therapeutic interventions [73]. Two blind reviewers independently evaluated all cases, and a third reviewer synthesized the scores, mediating discrepancies when needed. Case reports were assessed out of a total of eight points, while case series had a maximum of 10 points. Our criteria for inclusion demanded that case reports achieve a score of 6/8 and case series attain 8/10. This culminated in a quality threshold of 75–80%, deemed adequate for our analysis by the reviewing team [74].

Our systematic review scoured literature focusing on acute stroke patients, particularly emphasizing the correlation between stroke locations and types vis-à-vis delirium status. Our search criteria encompassed publications from January 2010 through June 2021, spanning multiple languages. We excluded case studies with fewer than 20 patients, case-control studies, and randomized controlled trials. Our search extended across various databases, including MEDLINE, EMBASE, PsycINFO, CINAHL, and Alois. We utilized either bivariate random effects models or network meta-analysis for determining pooled relative risks, while the methodological integrity was scrutinized across eight defined parameters [75]. Our endeavor culminated in the inclusion of 31 patient samples, representing a total of 8329 patients. We unearthed intriguing patterns—delirium was notably prevalent in patients with supratentorial as opposed to infratentorial lesions, anterior circulation compared to posterior, and cortical versus subcortical lesions [76]. Interestingly, the side of the stroke (right vs. left) did not exhibit any correlation with delirium. Delirium was also discernibly higher in patients suffering from hemorrhagic strokes relative to ischemic ones and those with pre-existing qualitative atrophy. This review has shed light on the intricate links between various brain regions, stroke types, and the onset of delirium. However, one must exercise caution in drawing conclusions due to the variability across studies and the sometimes vague descriptions of lesions. Nevertheless, these findings offer a pivotal foundation for predicting delirium risks in acute stroke scenarios and can pave the way for further studies probing into the neural circuits and pathological underpinnings contributing to delirium’s pathophysiology [75].

A multidisciplinary approach is becoming an increasingly integral part of healthcare, especially in areas like stroke care. Organized stroke care typically involves a dedicated team that is either stationed in a stroke ward, functions as a mobile team, or operates within a broader rehabilitation service [77]. This team is composed of various specialists, each bringing unique expertise to cater to the intricate needs of stroke patients. The effectiveness of this approach is substantiated by 23 trials that have demonstrated that, compared to other care methods, multidisciplinary stroke unit care significantly reduces mortality rates and dependency rates in patients during a median follow-up of one year [78].

The foundation of the observations in this research mainly rests on Cochrane and other systematic reviews post-2000, accompanied by other relevant quantitative and qualitative studies aimed at enhancing post-stroke recovery. One salient feature of stroke recovery is its complexity and individual variability, underscoring the critical role of healthcare professionals in collaborating efficiently. This synergy ensures that a gamut of collective knowledge and specialized skills is accessible for the betterment of stroke survivors [79].

There is a distinction between multidisciplinary and interdisciplinary functions, which is essential to understanding their contribution to stroke care. The patient journey post-stroke can be seen in three primary stages: the immediate response and emergency department phase, the inpatient care phase, and the post-discharge phase, which includes long-term support. At each of these junctures, multidisciplinary teams (MDTs) play distinct roles, shaping the trajectory of the patient’s recovery [79].

Policy framers and clinical guideline developers have consistently linked MDT practices to enhancements in the quality of stroke care [80]. The national stroke strategy for countries like England, Wales, and Northern Ireland earmarks a significant portion of its quality indicators towards MDT’s roles in ensuring effective service delivery and fostering improved patient outcomes. This faith in MDTs emanates partly from rigorous reviews of inpatient stroke care trials. Such reviews have unambiguously revealed the advantages of organized care in stroke units by MDTs, not just in reduced mortality but also in fostering patients’ independence, as evidenced by a higher number of patients living at home a year post-stroke [78].

Despite the clear advantages, the exact dynamics of how MDTs contribute to these improved outcomes have not been firmly determined. While the ideas behind multidisciplinary and interdisciplinary services in stroke care are widely accepted [80], it is imperative to distinguish between them and understand their unique impacts on post-stroke recovery.

When classifying interventions, the primary distinguishing factors are the method of delivery and the setting. For analytical purposes, outcomes from studies with similar methodologies were grouped. In cases where the same study population was reported in separate journals, they were treated as a singular entity for mortality assessment [81,82,83]. To address the diverse range of home-visit interventions, they were segmented based on the facilitator, either being led by a multidisciplinary team or other healthcare providers such as physiotherapists, nurses, or occupational therapists [84,85]. Further granularity in the analysis was achieved by categorizing interventions based on their nature and follow-up duration. A prominent metric used for assessment is the Barthel index (BI), a widely recognized tool to evaluate daily life activities (ADL). Different evaluation methods were discerned in the meta-analysis, offering a comprehensive view of post-intervention outcomes [86].

The economic implications associated with post-acute care (PAC) programs for stroke patients undergoing rehabilitation remain underexplored in the academic literature. This pioneering study delves into the economic burden shouldered by stroke patients engaged in PAC rehabilitation and gauges the effectiveness of multidisciplinary PAC programs in terms of both cost and patient functional outcomes [87]. Out of the 910 patients with stroke observed from March 2014 to October 2018, they were divided into two cohorts: those receiving PAC (from two medical centers) and those not involved in PAC (from three regional hospitals and a district hospital). This allocation was achieved through propensity score matching, maintaining a 1:1 ratio. To decipher the study’s economic aspects, a cost-illness framework was implemented, targeting specific cost categories. Remarkably, the direct medical costs for patients under the PAC system, which used a per-diem-based costing approach, were considerably lower than those in the non-PAC system, which utilized a fee-for-service approach (*p* < 0.001). The yearly economic load for stroke patients undergoing PAC rehabilitation stands around USD 354.3 million (based on 2019’s conversion rate of NT USD 30.5 to USD 1) [88]. Furthermore, functional improvement was notably more significant in the PAC cohort compared to the non-PAC cohort. This difference was both pre- and post-a year-long rehabilitation regime (*p* < 0.001). The emphasis on early rehabilitation following a stroke is evident, as it fosters health restoration, boosts confidence, and enhances the self-care abilities of patients. Evidently, PAC rehabilitation curtails the transitional period to the rehabilitation ward, showcasing its efficiency in cost-saving and functional improvement for stroke patients [89].

A structured, multidisciplinary team intervention immediately following an acute stroke can substantially reduce functional impairments, stave off complications, and subsequently lessen extended hospital stays. Data from the get with the guidelines–stroke (GWTG-Stroke) program reveals that of the 616,982 adults diagnosed with stroke, a staggering 90% were evaluated for potential rehabilitation during the acute phase [90]. Yet, a striking disparity emerges when we scrutinize inpatient stroke rehabilitation practices in Taiwan. Here, only 34.0% utilized rehabilitation services, which, when broken down, were 33.0% for physical therapy, 19.6% for occupational therapy, and a meager 5.3% for speech therapy. This is in stark contrast to the figures from countries like the United States, Canada, the UK, and Austria, where the utilization rates hover between 59% and 75% for physical therapy, 16% and 39% for occupational therapy, and 10% and 23% for speech therapy [91]. Another challenge in Taiwan is the delivery of inpatient rehabilitation, often restricted to bedside programs in certain local hospitals devoid of dedicated rehabilitation facilities. Additionally, a skills gap is apparent, with some local hospital therapists possessing limited experience in stroke rehabilitation. A potential remedy could be refining the payment system to encourage more skilled rehabilitation providers. The findings from our study reinforce the notion that initiating intensive stroke rehabilitation early leads to outcomes that are both cost-effective and efficient. The synergy of a multidisciplinary team is pivotal in realizing these outcomes [89].

The patient’s journey in managing strokes underscores the vital role of time. Indeed, when confronted with a stroke, every second matters; swift action can profoundly influence the effectiveness of interventions and significantly shape a patient’s ultimate health outcomes [92]. Regrettably, delayed hospital presentations are common, often sidelining potential treatments that could be beneficial if administered promptly. A critical impediment to swift hospital arrivals is the prevalent gap in awareness: both patients and bystanders often lack an understanding of the early warning signs and risk factors associated with stroke. This gap can manifest in a myriad of ways—from patients dismissing their symptoms, holding onto the hope that they will simply dissipate with time, to outright denial of the potential severity of their condition [93,94].

What is more concerning is the discrepancy in health awareness: stroke, despite its alarming prevalence and profound implications, lags behind in public consciousness. This lack of awareness is starkly evident even when juxtaposed with other grave conditions like acute coronary syndrome (ACS), cancer, or AIDS. Worryingly, even among those who have weathered a stroke episode, a significant portion remain inadequately informed about the disease [95].

To bridge this awareness gap, several public health initiatives and campaigns have been launched. Slogans such as “Stroke Chain of Survival”, “Time is Brain”, and “Face, Arms, Speech, Time (FAST)” aim to hammer home the urgency of stroke and the importance of early medical intervention [96,97,98]. The underlying hypothesis of these campaigns is straightforward: arming the public with knowledge about stroke’s warning signs can expedite the decision to summon emergency medical services, a decision that could prove life-saving [99].

Furthermore, cognizance of risk factors is not solely about timely interventions. Comprehensive awareness can also galvanize primary and secondary preventative measures. By acquainting individuals with the risks, there is an opportunity to inspire preventative lifestyle changes, which could significantly curtail future cerebrovascular issues. However, the current reality is sobering: a substantial segment of individuals at elevated risk for strokes remains in the dark about their perilous position [100]. This underscores the urgent need for continued and intensified educational outreach to the broader population.

Awareness about stroke among the general populace is pivotal. Its presence or absence can spell the difference between timely intervention for an individual suffering an acute stroke and potential delayed treatment or mismanagement [101]. This study delves into this awareness in the Silesian voivodeship, the most densely populated region of Poland. The goal was not only to gauge the extent of general knowledge about stroke but also to discern “adequate knowledge of stroke”, which is a comprehensive understanding that encompasses risk factors, symptoms, and the necessary actions to take when confronted with an acute stroke. This holistic understanding is imperative for efficacious stroke management. Furthermore, pinpointing the factors that influence this adequate knowledge can be instrumental in tailoring educational strategies [102].

A custom survey has been employed, all pertinent to stroke. Beyond just querying their understanding of individual stroke aspects, we looked into their “adequate knowledge of stroke”, which encapsulates their understanding of risk factors, the recognition of symptoms, and the protocol to follow during an acute stroke event.

Out of those surveyed, 834 individuals (73.5%) could accurately define what a stroke is. An encouraging 92.8% recognized a stroke as a medical emergency, with 97.5% acknowledging the need for medical intervention [103]. On the flip side, a concerning 42.4% could not pinpoint a specific stroke symptom, and only a mere 38.6% could enumerate at least two or more risk factors. This culminated in just 36.3% of the respondents having what we classify as adequate knowledge of stroke. Upon analysis, factors like the length of education, a personal connection to someone who suffered a stroke, gender, and place of residence emerged as key determinants of this comprehensive stroke knowledge [104].

The populace of southern Poland displays a level of stroke awareness that can be deemed inadequate, especially when considering the urgency and timeliness needed for effective stroke management.Personal experiences, particularly having a friend or relative who has suffered a stroke, stand out as the most influential factor in having adequate stroke knowledge [105].

The role of emergency medical services (EMS) cannot be overstated in the realm of stroke management. By mitigating pre-hospital delays and ensuring swift in-hospital assessments, they hold the potential to be game-changers. Critical steps like recognizing stroke symptoms at the dispatch center, ensuring on-the-spot stroke diagnosis and triage, expediting patient transfer to equipped facilities, and notifying hospitals in advance can dramatically reduce pre-hospital lags. These steps not only amplify the chances of administering rtPA but are imperative for timely acute stroke care in our country [106].

One major limitation is its confinement to a singular stroke center, which poses questions about the wider applicability of our findings on a national scale. While the results are telling, the reality in other regions might be even more grim, given the paucity of stroke awareness drives, the lack of specific pre-hospital protocols for stroke, and the expansive patient base covered by many county hospitals [107]. Additionally, it was earmarked the stroke onset as the time a patient was last observed without symptoms, neglecting the moment they were first spotted with them. Moreover, it did not delve into the actions of bystanders, which could significantly impact the decision to summon EMS. Addressing these lacunae will be pivotal in subsequent studies, especially if we aim to truly discern the barriers preventing stroke patients in Romania from accessing IVT [108].

## 3. Neuroimaging and Neuroradiology: A Deep Dive

### 3.1. The Evolution of Neuroimaging in Stroke Diagnosis and Its Historical Context

The domain of neuroimaging has witnessed considerable evolution over time, playing a transformative role in the diagnosis and management of stroke, specifically in relation to RSSI in patients displaying lacunar syndromes. A stroke, simply put, is a form of injury to the brain due to interrupted blood flow, and early detection is crucial for timely and effective intervention.

To diagnose an RSSI, one needs to pinpoint lesions through neuroimaging like CT or MRI scans, which are consistent with a minor ischemic stroke in specific regions of the brain. These regions include the area served by particular deep perforating arteries such as lenticulostriate and thalamoperforating, among others. This would involve the subcortical white matter areas (like the centrum semiovale and corona radiate, to name a few) or deep gray structures, including the basal ganglia and nuclei located in the brainstem [109].

Historically, CT scans emerged as the pioneering neuroimaging technique that could discern small focal hypoattenuations, synonymous with lacunar strokes. Nevertheless, during the nascent hours following the symptom onset, CT scans found it challenging to highlight these tiny subcortical infarcts, often confusing them with pre-existing lesions in patients grappling with SVD [109].

The advent of MRI revolutionized stroke diagnostics. It facilitated nuanced morphological and topographical characterization of RSSI, offering a more detailed snapshot of the brain’s structures [110]. The implementation of diffusion-weighted imaging (DWI) within MRI proved to be a game-changer. It enabled the identification of recent ischemic changes by displaying hyperintensities mere minutes after the onset of a stroke, which could remain visible for roughly 3 to 5 weeks. Meanwhile, older lesions could be discerned through other structural sequences in the MRI [111,112]. However, MRI, despite its precision, is not foolproof. Factors like the magnetic field strength, motion artifact correction, and the sequencing of image acquisition can sometimes lead to small lesions going undetected [113]. Hence, even if DWI does not highlight hyperintense lesions, one should not hastily dismiss the possibility of a lacunar stroke, especially if the patient’s symptoms suggest otherwise.

Additionally, an RSSI evident on an MRI might actually be the aftermath of a larger deficit in blood flow, possibly affecting a larger territory than initially thought, as some perfusion studies have shown [114]. On occasion, perfusion deficits linked to a single perforating artery might be reversible. It is essential to remember that stroke is a complex and dynamic process, influenced by a plethora of factors like metabolic demand, time of ischemia, and collateral blood supply, to name a few. Hence, while imaging provides invaluable insights, it may not capture the entire narrative, necessitating a comprehensive clinical assessment and additional tests for a holistic understanding [115].

In a study involving 312 stroke patients who underwent CT scans, 37 displayed clinical signs of lacunar syndrome. Of these, 18 exhibited lacunar-sized infarcts on their scans, 13 had unremarkable scans, and 6 surprisingly revealed large infarcts. Intriguingly, of these six patients, five manifested pure motor hemiplegia, and one exhibited a pure sensory stroke. Both clinical evaluations and angiography unveiled potential treatable sources of emboli in both lacunar-sized and large infarcts [116]. This leads to two pivotal conclusions:A clinical lacunar syndrome does not always correlate with the size of the infarct—it can sometimes be linked to a larger infarct.Identifying a lacunar infarct through a CT scan does not negate the need for further angiographic studies, especially if there is a likelihood of detecting an embolic source [109].

### 3.2. An In-Depth Look into Neuroimaging Modalities and Their Role in Post-Stroke Recovery Prediction

Neuroimaging serves as an indispensable tool in the realm of stroke diagnosis and management. Beyond its foundational role in distinguishing ischemic strokes from hemorrhagic strokes in the acute phase, neuroimaging has been emerging as a pivotal component in decision-making for cutting-edge treatments, such as late-window thrombectomy. In this review, our focus extends beyond immediate diagnostic applications, shedding light on the potential of neuroimaging techniques in forecasting post-stroke recovery.

Integrating recovery predictions with quantifiable measurements permits the identification and development of biomarkers, which can be monumental in the treatment and management of stroke patients. As defined by the FDA-NIH biomarker working group, a biomarker is a “specifically identified metric serving as an indicator of natural biological activities, pathological processes, or responses to an intervention or treatment” [117]. This definition underscores the importance of unifying terminology across scientific disciplines. Over time, as our understanding of stroke has deepened, the term ‘biomarker’ has evolved. It has transitioned from being primarily a diagnostic tool to being intricately linked with therapeutic mechanisms. Presently, biomarkers encompass a broad spectrum of factors, ranging from genetic markers and molecular indicators to clinical scales and, crucially, neuroimaging and neurophysiological indicators.

Stroke recovery biomarkers sourced from neuroimaging encapsulate both structural and functional dimensions [118]. For structural evaluation, parameters such as the size of the infarct, the degree of cortical or white matter damage, the integrity of white matter, and the percentage of injury to the corticospinal tract are of paramount importance. On the other hand, functional evaluations hinge on aspects such as activation patterns within ipsilesional (same side of the brain as the lesion) and contralesional (opposite side to the lesion) regions, the balance between the hemispheres, connectivity during resting states, synchronization and desynchronization during specific tasks, and measures of cortical excitability, facilitation, and inhibition [119].

Following this, we delve into a succinct discussion on specialized techniques tailored for analyzing stroke recovery. This exploration is particularly pertinent when considering the intricate processes of angiogenesis (formation of new blood vessels) and neuroplasticity (the brain’s ability to reorganize and adapt). The nuances of these techniques, offering insights into their capabilities and unique characteristics, are consolidated in Table 1 for ease of reference [120]. This table serves as a valuable resource, offering readers a comprehensive overview of the breadth and depth of neuroimaging modalities available in contemporary stroke research and recovery prediction.

### 3.3. Post-Stroke Angiogenesis and the Expanding Horizons of Advanced Neuroimaging

Angiogenesis following a stroke is an intricate, multi-phased procedure. It begins with gene transcription and the release of proangiogenic factors, leading to a cascade of events including the proliferation of endothelial cells and the sprouting of new vascular structures, culminating in the formation of microvessels [121]. Today’s imaging methodologies can investigate an array of both structural and functional characteristics within tissues [122]. Breakthroughs in magnetic resonance imaging (MRI) have ushered in techniques capable of assessing tissue blood flow and deducing various metrics related to the vascular network, such as microvascular cerebral blood volume (CBV) and the density of these microvessels [123].

Highlighting these advancements, an experimental study conducted by Yanev et al. utilized steady-state contrast-enhanced (ssCE-) MRI with an extended blood pool circulation time to delineate vascular changes within ischemic lesions and associated regions, spanning from the subacute to the chronic stages post-cerebral stroke [124,125]. Their findings elucidated dynamic vascular regeneration in areas surrounding the lesion and ongoing neovascularization in areas linked to but not directly impacted by ischemia [126,127,128]. Such vascular activities could play pivotal roles in the repair and restructuring of non-neuronal tissues, influencing post-stroke recovery dynamics. The nascent stages of angiogenesis can be detected via MRI techniques as disruptions in the blood–brain barrier [129,130]. This disruption, or permeability, correlates with the proliferation of endothelial cells and the initiation of vascular sprouting. To detail the integrity of the blood–brain barrier, dynamic contrast-enhanced MRI (DCE-MRI) leveraging gadolinium chelates can be harnessed, especially when alterations in MRI signals arise due to contrast seepage into surrounding tissues [131,132,133].

In the broader realm of stroke management, cutting-edge neuroimaging stands as a vital asset, aiding clinicians in bypassing the time restrictions and expanding the application scope of intravenous thrombolysis (IVT) [134,135,136]. To understand the potential effects of employing advanced neuroimaging (AN), specifically CT/MR perfusion, on the outcomes of acute ischemic stroke (AIS) patients undergoing IVT, irrespective of the elapsed time since symptom manifestation [136,137]. Through a retrospective lens, we analyzed AIS patients who underwent IVT as a sole therapeutic intervention over a span of six years. Our focus was on discerning if there were notable differences between patients who had undergone advanced neuroimaging prior to IVT (AN+) versus those who had not (AN−). Key outcome metrics ranged from clinical safety indicators, such as intracranial hemorrhage and 3-month mortality, to efficacy measures like door-to-needle time, discharge neurological status (NIHSS-score), and 3-month functional status gauged by the modified Rankin Scale (mRS) [138]. Interestingly, while the utilization of IVT monotherapy saw an uptick in the AN+ cohort, the key metrics across both groups remained comparable, suggesting the AN+ approach does not compromise the efficacy or safety of IVT treatment.

To summarize, our exploratory study underpins the value of integrating advanced neuroimaging into the acute stroke treatment pathway for AIS patients. Not only does it augment the administration rate of IVT, but it also maintains the treatment’s efficacy and safety profile, offering promising avenues for enhanced patient care [139,140].

### 3.4. Advancements in Stroke Treatment and the Role of Neuroimaging

In recent years, the landscape of stroke treatment has witnessed transformative progress. A significant stride forward was marked by the DAWN and DEFUSE 3 trials in 2018. These groundbreaking studies unveiled the effectiveness of mechanical thrombectomy beyond the conventional 6-h timeframe, extending the treatment window up to 24 h post-onset of acute stroke symptoms in patients with large vessel occlusions (LVO) [141]. This paradigm shift was rooted in judicious patient selection, emphasizing a mismatch between the infarcted core and the surrounding at-risk, yet salvageable, ischemic penumbra as depicted in perfusion images. Essentially, reperfusion treatments aim to rescue the endangered penumbra and forestall the expansion of the infarct core [142].

Highlighting the trials, the DAWN (DWI or CTP assessment with clinical mismatch in the triage of wake-up and late-presenting strokes undergoing neurointervention with trevo) trial stands out as a multi-center, randomized controlled investigation. It focused on patients presenting 6 to 24 h after the emergence of stroke symptoms and exhibiting a proximal LVO. Enrollment was based on detecting a mismatch between the identified ischemic core via DWI or CT perfusion and the degree of neurological impairment (manifested as an NIHSS score of 10 or above). The median interval between symptom onset and intervention was found to be 12.5 h. Notably, the outcomes illustrated a pronounced improvement in patients undergoing mechanical thrombectomy compared to conventional treatments: 49% of these patients exhibited minimal to no disability, a stark contrast to the 13% in the standard therapy cohort [143].

In specialized stroke centers, comprehensive clinical evaluations coupled with advanced neuroimaging techniques are routinely employed, paving the way for predictive assessments of patient trajectories. A wealth of literature delves into the interplay of clinical and neuroimaging measures, with a special emphasis on proprioception during the subacute post-stroke phase [144,145]. Notably, clinical indicators, including attentional capacities and daily functioning metrics like the behavioral inattention test (BIT) and the functional independence measure (FIM), have displayed strong correlations with proprioceptive assessments [146,147]. Neuroimaging dimensions, such as lesion volume and precise regional damage, have further enriched our understanding, linking larger lesions with deteriorated post-stroke proprioceptive outcomes [148,149]. Cutting-edge tools like voxel-based lesion-symptom mapping (VLSM) have allowed researchers to discern the statistical interrelations between affected brain areas and post-stroke proprioceptive capacities [150,151]. However, while motor recovery has been extensively studied to identify early recovery predictors, research focused on forecasting long-term proprioceptive recovery remains relatively sparse [118].

Against this backdrop, our study set out to bridge this knowledge gap, seeking to ascertain the predictive prowess of combined clinical, neuroimaging, and robotic evaluations in forecasting long-term proprioceptive outcomes. The study’s central objective was to gauge the efficacy of these measures, gathered within the initial fortnight post-stroke, in forecasting proprioceptive deficits at the six-month mark [152]. Drawing from prior correlations between clinical metrics, neuroimaging results, and proprioceptive evaluations, we hypothesized that standalone clinical or neuroimaging models would offer satisfactory predictive accuracy for six-month proprioceptive outcomes [153]. Expanding on this, we further postulated that robotic metrics, either in isolation or synergized with clinical and neuroimaging data, would enhance predictive accuracy and bolster the area under the receiver-operator characteristic (ROC) curve (AUC) [154].

### 3.5. Understanding Cerebral Artery Occlusions and the Evolution of Stroke Treatments

In the realm of stroke care, a deeper understanding of cerebral artery occlusions and their clinical implications is pivotal. The year 2015 marked a significant breakthrough in this domain. Groundbreaking trials, namely MR CLEAN, ESCAPE, SWIFT PRIME, REVASCAT, and EXTEND IA, conclusively demonstrated the superiority of endovascular thrombectomy over standard medical management in treating anterior circulation large vessel occlusion strokes [155]. The potency of endovascular thrombectomy as a treatment modality is evident, with a patient ‘number needed to treat’ ranging from a mere 3 to 10. A set of criteria, including occlusion location (focusing on proximal anterior occlusions such as the internal carotid or middle cerebral artery), time since the onset of the stroke (ideally within an early window of 6–12 h), and an acceptable level of infarct burden (reflected by an Alberta Stroke Program Early CT Score [ASPECTS] of ≥6 or an infarct volume of less than 50 mL), became the basis of patient selection for these trials [156].

Subsequent trials in 2017, notably DAWN and DEFUSE-3, pushed the boundaries even further by successfully expanding the treatment window up to 24 h for a certain subset of patients. This paradigm shift has been embraced by societal and national thrombectomy guidelines, granting a Class 1A recommendation for the carefully selected patient cohort. However, the journey is far from over. Currently, randomized controlled trials are underway to study thrombectomy’s applicability in stroke subpopulations previously considered ineligible. These trials are fueled by promising insights from an aggregated analysis of early trials (by the HERMES collaboration) and budding retrospective data. The focal points of these trials include patients with large vessel occlusion strokes exhibiting mild deficits (with a national institutes of health stroke scale score less than 6) or those with a substantial infarct burden (a core volume exceeding 70 mL) [157].

On a global scale, stroke stands as the second primary cause of mortality. Given the limited therapeutic arsenal against ischemic stroke, there is an urgent need to innovate and expand treatment options. In recent years, metformin, primarily known for its anti-inflammatory properties, has been spotlighted for its potential neuroprotective capabilities against ischemic damage caused by the stop and restart of blood flow (ischemia/reperfusion) [158,159]. This study ventured to delve deeper into metformin’s efficacy, specifically in the context of permanent middle cerebral artery occlusion (pMCAO) without any subsequent reperfusion in rat models. Neurological aftermaths post-pMCAO were gauged using the Longa scale, a reliable metric for body movement evaluation. Additionally, the extent of brain damage and swelling were ascertained through the 2,3,5-triphenyltetrazolium chloride staining technique.

Diving into the findings, metformin administration led to marked neurological improvement and a decrease in infarct size, especially 120 h post-pMCAO. While metformin prevented neuronal loss in the ischemic cortex, its effect was not as pronounced in the striatum 48 h after pMCAO. An encouraging observation was the substantial decline in the number of total and activated microglia 48 h post-stroke upon metformin treatment. This anti-inflammatory action of metformin corresponded with a surge in interleukin 10 (IL-10) production 48 h following pMCAO. Cumulatively, this study furnishes compelling evidence for metformin’s anti-inflammatory and neuroprotective roles in a pMCAO setting [160].

### 3.6. Harnessing Neuroradiology for Therapy Planning: Delving into Techniques and Implications

Neuroradiology stands as a pillar for therapeutic planning, especially in the intricate landscape of neurological disorders. The path to decision-making through neuroradiology is a layered one, often blending advanced technology with human acumen.

Brain magnetic resonance imaging (MRI), for instance, is instrumental in prognosticating the clinical trajectory of patients with acute ischemic stroke (AIS) [161]. In recent years, there has been a technological windfall with deep learning (DL) techniques successfully employing brain MRI images and certain biomarkers for forecasting unfavorable outcomes in AIS patients. However, an intriguing dimension that has hitherto remained unexplored is the potential of using natural language processing (NLP)-oriented machine learning (ML) algorithms. The key distinction here is the source of data: free-text reports of AIS patients derived from brain MRI scans [161].

To chart this unexplored terrain, a study was conducted focusing exclusively on English MRI reports obtained during the admission phase for AIS patients. Defining poor outcomes as a modified Rankin scale score ranging between 3 and 6, data acquisition was meticulously overseen by a team of trained healthcare professionals. The emphasis was placed on the first MRI report obtained during hospitalization. Structuring the study, the collected text dataset was systematically segmented into training and testing batches, following a 70:30 proportion.

The data underwent three levels of vectorization: word, sentence, and document levels. The nuanced “bag-of-words” model found its application at the word level, which disregarded word sequence but tallied text token repetitions. On the other hand, the “sent2vec” methodology, which took the sequence of words into account, was employed at the sentence level. Meanwhile, word embedding was applied at the document level. Alongside traditional ML algorithms, DL paradigms like the convolutional neural network (CNN), long short-term memory, and multilayer perceptron were adopted. This ensemble was evaluated against 5-fold cross-validation and grid search techniques. A consistent performance metric, the area under the receiver operating characteristic (AUROC) curve, was employed.

Analyses from 1840 AIS subjects revealed a stark reality—a hefty 35.1% grappled with poor outcomes three months post-stroke onset. The random forest emerged as the top classifier at the word-level approach with an AUROC of 0.782. However, on a broader spectrum, the document-level approach eclipsed the other two. The multi-CNN algorithm set the gold standard in classification with an AUROC of 0.805, closely trailed by the CNN algorithm at 0.799. The crux of these findings lies in the supremacy of DL algorithms, particularly in NLP-based predictions, where multi-CNN and CNN outperform other neural networks in forecasting adverse outcomes [162]. This asserts the pivotal role NLP-fueled DL can play, marking its ascendancy as a digital beacon for unstructured healthcare data predictions [163].

Transitioning from MRI to the realm of computed tomography (CT), the trio of CT, CT angiography (CTA), and CT perfusion (CTP) reign supreme in emergency departments when there is a hint of cerebrovascular compromise [164]. The pressure-cooker environment of emergency settings demands precise and rapid image interpretation. Notably, the clinical tableau of an acute stroke can be mimicked by myriad conditions. Hence, the rapid discernment of true strokes from their imitators is paramount for clinicians. With a vast array of conditions masquerading as acute strokes, the onus falls on imaging to discriminate. While some of these conditions reveal themselves quite clearly, others can pose diagnostic challenges. Pictorial representations derived from CTP serve as vital clues [165]. A series of acute stroke instances and their look-alikes were presented, emphasizing the indispensable nature of these imaging “pictograms” for radiologists. This visual encyclopedia should bolster radiologists’ diagnostic prowess, ensuring they remain conversant with the nuances of diverse imaging techniques, reaping their advantages while steering clear of potential pitfalls [166].

### 3.7. Pioneering Neuroimaging Optimization: Charting the Path to Precision Diagnostics

The vast realm of ischemic stroke, which encompasses over 80% of all stroke occurrences, stands as a formidable adversary in the global health arena, frequently leading to mortality and long-term disabilities [167]. Administering recombinant tissue plasminogen activator (rt-PA) intravenously is an accredited countermeasure for acute ischemic strokes caused by larger arteries, provided it is employed within a 4.5-h window from the onset. Moreover, mechanical thrombectomy can serve as an intervention for large artery occlusions up to 24 h post-onset [167]. Yet, the real-world challenge lies in optimizing diagnostic processes for acute treatments. Factors such as minimizing the onset-to-needle time duration, ensuring rapid access to angioCT images, and facilitating timely magnetic resonance imaging (MRI) become bottlenecks in many healthcare frameworks. Such constraints often lead to a disparity between real-world prognosis rates and those recorded in randomized controlled trials. Notwithstanding the efficacy of applied endovascular techniques, there remains a lack of comprehensive understanding regarding certain cellular mechanisms post-reperfusion [168]. Furthermore, research areas like the changes in mitochondrial morphology and function related to reperfusion and ischemia-induced neuronal death remain relatively uncharted. A future vision in stroke research mandates an in-depth exploration of the evolving landscape of imaging techniques. It is crucial to comprehend the intricate relationship between the ever-refining imaging methodologies and factors like clot structure variability, vascular permeability, and the diverse manifestations of ischemic reperfusion damages, especially in the penumbra. Insights into these domains hold the key to devising targeted interventions that confer lasting health benefits [169].

### 3.8. Radiologists and Neurologists: Crafting a Symbiotic Diagnostic Journey

Seamless collaborations between radiologists and neurologists stand as a beacon of hope for patients. In our study, we meticulously screened patients using CT/CTA or MRI before initiating any intervention [170]. The focus group comprised acute ischemic stroke patients, primarily attributed to large vessel occlusions (LVO) and, more specifically, those at the M2 level. LVO categorizations included occlusions of the internal carotid artery (ICA), middle cerebral artery (MCA) at the M1 segment, intracranial vertebral artery (VA), and basilar artery (BA) as identified in CTA [171]. Our inclusion criteria encompassed: patients above 18 years of age, those with a national institutes of health stroke scale (NIHSS) score of 6 or higher (or presenting isolated aphasia), individuals who demonstrated prior functional independence using the modified ranking scale (mRS ≤ 2), and patients who sought medical attention within 6 h of stroke onset. Interestingly, our analysis also welcomed “wake-up” stroke patients who presented between 6 h and 24 h post-onset but exhibited a discernible mismatch between ischemic core and penumbra as per MRI readings. For all qualifying patients, intravenous thrombolysis (IVT) using rt-PA was the first line of intervention. If IVT was contraindicated, the subsequent step was mechanical thrombectomy post-CT and CTA evaluation. Crucially, intervention decisions were a collective resolution made by a cohesive team of neurologists, radiologists, and interventionists. The exclusions were patients displaying pronounced massive strokes on scans, especially encompassing more than a third of the MCA’s territory. Ethical considerations were paramount, with informed consent being a requisite for all study participants [170,171].

### 3.9. Leveraging Deep Learning in Neuroimaging: A Paradigm Shift

Neuroimaging, especially brain CT scans, forms the bedrock of cerebral evaluations. Yet, the intricate task of interpreting emergent brain CT findings demands a high degree of expertise and can be labor-intensive for even adept neuroradiologists. Herein lies the prowess of deep learning, especially convolutional neural networks (CNN), which have been revolutionizing the medical imaging landscape [172]. Our study proposed the utilization of CNN-centric deep learning paradigms to efficaciously categorize strokes based on unenhanced brain CT imagery into normal, hemorrhage, infarction, and other diverse categories. The models under our analytical radar were CNN-2, VGG-16, and ResNet-50. These were not naive models; they had undergone prior training via transfer learning, adapting to various data magnitudes, mini-batch dimensions, and optimization algorithms. Their efficacy was put to the test with brain CT images. The findings were enlightening: when juxtaposed against other research outcomes, our models, especially CNN-2 and ResNet-50, showcased superior performance. Notably, while ResNet-50 clinched an impressive accuracy score of 0.9872, it took a tad longer to render outcomes in comparison to its counterparts. In essence, with the right hyperparameter fine-tuning, these deep learning models can be pivotal in clinical scenarios, aiding neurologists and radiologists in determining potential hemorrhagic strokes, infarctions, or other neurological manifestations [173].

Functional neuroimaging has significantly furthered our grasp of neural processes involved in post-stroke recovery and enhancements derived from brain stimulation. The variability observed among individuals in terms of recovery and response to treatment can be associated with imaging markers, notably connectivity. Cutting-edge methods in fMRI data analysis, like dynamic functional connectivity, enable exploration of stroke-induced changes in temporal network dynamics and their relationship to motor deficits. Nevertheless, we are yet to achieve a tailored approach that accounts for unique network pathology to accurately rectify specific network node dysconnectivity. Preliminary efforts, such as employing multivariate machine learning approaches to forecast motor deficits or outcomes based on initial post-stroke fMRI data, have been undertaken. Still, the validity of relying solely on a single MRI network marker for individualized predictions to ensure diagnostic precision remains a topic of discussion [172].

### 3.10. Comprehensive Cerebral Imaging: Collaborative Diagnostics in Emergency Care

Upon their admission to the emergency department, every patient underwent a cerebral computed tomography (CT) scan, which was conducted either with or without the administration of a contrast agent. This immediate imaging step was paramount to ensuring a swift diagnostic process [174]. The precise nature, severity, and anatomical location of the stroke were then meticulously diagnosed. This pivotal task was a collaborative effort between two medical experts: the radiologist who executed the brain imaging and the neurologist who conducted the clinical evaluation of the patient. This dual expertise ensured a comprehensive understanding of the patient’s neurological condition, fostering more informed clinical decisions [175].

For a definitive diagnosis of acute ischemic stroke (AIS), the medical team adhered to the World Health Organization’s long-standing definition of a stroke. Introduced back in 1970 and still deemed relevant in modern clinical practice, this definition characterizes a stroke as a “sudden manifestation of clinical symptoms pointing towards a focal (or occasionally global) disruption of cerebral functions. These symptoms persist for a duration exceeding 24 h or may even culminate in the patient’s death. Notably, the only discernible causative factor for these symptoms should be of vascular origin, unless there are specific interventions like surgery or medication that might interrupt this course” [176]. This clear-cut definition, along with the combined insights from imaging and clinical examinations, solidifies the diagnostic accuracy, ensuring patients receive the most appropriate and timely care.

## 4. Treatment Paradigms

### 4.1. Pharmacological Approach

#### 4.1.1. Exploring Intravenous Thrombolytic Agents: An In-depth Analysis of Mechanisms, Advantages, and Potential Hazards

Alteplase and Potential Alternatives in Stroke Treatment. Alteplase currently stands as the sole drug greenlighted by the FDA for the thrombolysis of acute ischemic stroke (AIS). However, the research horizon is dotted with other thrombolytic agents that might potentially rival or replace alteplase in the future. This comprehensive study dives deep into the potency and safety of an array of such agents—urokinase, ateplase, tenecteplase, and reteplase. Through sophisticated computational simulations entwining both pharmacokinetics and pharmacodynamics, paired with a meticulous local fibrinolysis model, we benchmarked the drugs against multiple metrics: clot dissolution timeframe, resistance to plasminogen activator inhibitor (PAI), potential risk of intracranial hemorrhage (ICH), and the latency from drug introduction to clot dissolution [177].

Our insights highlighted that urokinase boasted the most rapid clot dissolution. However, it also carried an elevated risk of ICH, linked to its propensity for excessive fibrinogen depletion in the plasma. Tenecteplase and alteplase showcased akin efficiencies in clot dissolution, but the former exhibited a reduced ICH risk and demonstrated superior resilience against PAI-1. Reteplase, interestingly, showed the most sluggish rate of fibrinolysis but left fibrinogen concentrations in systemic plasma untouched. To comprehend these intricacies further, we utilized a 1D mathematical model from prior research, offering predictions on the therapeutic results of these drugs based on their inherent properties and modes of action. The model reinforced clinical observations, advocating the possible supremacy of tenecteplase over alteplase while underscoring reteplase’s limited efficacy despite its diminished ICH risk [178,179,180]. As AIS clinical studies inherently bear unpredictable hazards, computational models like ours could be pivotal in fine-tuning dosage regimens for multi-drug therapies or newly conceptualized drugs, given a clear understanding of their PKPD mechanisms and kinetic reactions [181].

2.Intracerebral Hemorrhage’s Impact on Mortality Rates. Taking a retrospective stance, this study sifted through data from South Korea’s national health insurance service database, spanning 2005–2018. The focal cohort consisted of hyperacute ischemic stroke patients who had undergone intravenous thrombolysis. A stark comparison was drawn between ICH-afflicted patients and those who avoided ICH. An alarming revelation was that within the 12-month window post-treatment, the mortality rate in the ICH cohort was more than double that of their counterparts (42.8% vs. 17.5%). This suggests that ICH post-thrombolysis can drastically heighten the risk of mortality in hyperacute ischemic stroke patients, amplifying it nearly threefold [182,183].3.The Promise of Plasmin Nanoformulations in Ischemic Stroke Treatment. While thrombolytic therapy remains the gold standard for treating ischemic strokes, its current mainstream agent, the tissue plasminogen activator (tPA), often encounters obstacles due to an associated hemorrhage risk. Plasmin, a direct fibrinolytic agent, offers a safer hemostatic profile. However, its therapeutic potential diminishes when introduced intravenously due to rapid inactivation by anti-plasmin. To navigate this, nanoformulations have emerged as viable tools to enhance drug stability. This study unveils a groundbreaking nanoformulation for plasmin, demonstrating increased stability and heightened therapeutic efficacy, potentially redefining ischemic stroke treatment [184].4.Decoding Blood–Brain Barrier Deficits Post-Stroke. Utilizing advanced two-photon microscopy, Knowland et al. have painted a vivid picture of changes in tight junctions (TJs) post-stroke in transgenic mouse models. Observations indicated that the blood–brain barrier (BBB) started leaking as early as 6 h post-ischemia, even though profound structural defects in TJs only became evident after 48 h. The increase in endothelial caveolae and transcytosis rate post-ischemia suggests a sequential deterioration of barrier mechanisms, highlighting the multifaceted impacts on the BBB following a stroke [185].

Top of Form

Under normal physiological conditions, the blood–brain barrier (BBB) stands as a sentinel at the frontier of the central nervous system (CNS), exercising selective permeability to maintain a stable environment. A couple of primary features define this selective permeability. First, endothelial cells (ECs) in the CNS are notable for their sparse vesicular population and an almost nonexistent rate of transcytosis. Second, there exists a sophisticated system of tight junctions (TJs) that bridge these ECs, creating an almost impenetrable seal [186].

A protein of considerable significance in this scenario is the major facilitator superfamily domain containing 2a (Mfsd2a). This protein has the unique distinction of being predominantly expressed in blood vessels that harbor the BBB. Intriguingly, its expression and activity are orchestrated by surrounding pericytes. Mfsd2a does not merely exist as a passive structural element; it plays a proactive role in crafting a specific lipid milieu. This environment is crucial as it actively inhibits the formation of caveolae vesicles within CNS ECs. By doing so, it curtails transcytosis, bolstering the integrity of the BBB [187].

However, during acute ischemic stroke (IS), this meticulously balanced environment faces upheaval. The aftermath of ischemia, as early as 6 h post-event, witnesses a surge in the number of endothelial caveolae. This spike is not merely quantitative; it ushers in an escalated rate of caveolae-mediated transcytosis. Macromolecules, which under regular circumstances would be barred entry, like albumin, now find passage via this heightened caveolae-mediated transcytosis. Meanwhile, the trusted sentinels of the CNS, the tight junctions, remain largely uncompromised initially. It is only after the 48-h mark post-ischemia that these TJs betray glaring structural anomalies [188]. These cellular-level dynamics do not occur in isolation. They are accompanied by collateral shifts in the surrounding CNS architecture. The surge in CNS endothelial vesicles coincides with discernible alterations in the pericytes’ basement membrane coverage. This vascular restructuring is mirrored by astrocytic responses. The usually slender astrocytic end-feet swell, displaying signs of stress. This edema is further echoed in their mitochondrial structures, which too show signs of swelling and possible dysfunction [189]. In sum, acute ischemic stroke initiates a cascade of events that disrupt the once-steadfast BBB, compromising its selective permeability and potentially influencing disease progression and recovery (Figure 1).

#### 4.1.2. Fibrinolytic Therapy in Acute Ischemic Stroke: A Comprehensive Analysis

##### Understanding the Fibrinolytic Approach and its Clinical Applications

Thrombolytic therapy, aimed at breaking up clots that obstruct blood flow, is critical in the management of conditions such as acute ischemic stroke (AIS) and acute myocardial infarction (AMI). However, this treatment is not without its challenges. A prominent concern is the risk of harmful hemorrhagic complications, prompting clinicians to establish a specific time window within which this therapy can be administered with optimal safety [186]. Efforts are currently underway, through various basic and clinical studies, to innovate next-generation thrombolytic drugs that might potentially broaden this time window.

In AMI-focused clinical trials, a wealth of pharmacokinetic and pharmacodynamic (PKPD) data, like temporal plasma concentrations of tissue plasminogen activator (tPA) and other fibrinolytic proteins (e.g., fibrinogen, plasminogen, and α2-antiplasmin), have been harvested. Such data are invaluable for deepening our understanding of fibrinolysis mechanisms and discerning the impact of dosage regimens on both therapeutic efficacy and potential toxicity [187,188,189]. In contrast, AIS clinical trials have predominantly concentrated on clinical and neurological outcomes, such as those measured by the modified Rankin scale and NIHSS. This disparity in focus has posed challenges. Given the inherent pathological distinctions between AMI and AIS, there is an observed variance in response to tPA dosage. Consequently, it becomes challenging to transpose the PKPD insights gleaned from AMI trials directly onto AIS patients [190,191]. Notably, due to the inherent risks associated with AIS, such as a heightened propensity for intracranial hemorrhage (ICH) linked to acute brain infarction, lower tPA dosages are recommended for AIS compared to AMI. This dosage recommendation is primarily anchored in clinical outcomes rather than PKPD studies, amplifying the call for more PKPD research in AIS thrombolytic therapy. A practical solution to this dilemma might be the deployment of mathematical models, especially when real-time clinical PKPD studies on stroke patients are impractical given the time-sensitive nature of the treatment. For instance, we have pioneered a computationally adept simulation platform for AIS thrombolysis. This model, accounting for systemic drug effects and the 1D progression of clot dissolution, can predict outcomes like lysis completion time and plasma FBG levels, indicative of ICH risk, without cumbersome computational demands. This model’s versatility is evident as it can emulate various therapeutic scenarios and can be tweaked to mirror different fibrinolytic drugs besides alteplase [192]. Furthermore, it can be adapted to diverse AIS conditions by adjusting local PD model parameters and clot features anchored in patient-specific data. Future ambitions for this model include expanding its scope to scrutinize the influences of diverse therapeutic and physiological factors on recanalization rates and to craft bespoke dosing regimens through model-driven optimization [193].

At the root of ischemic strokes is an artery blockage in the brain, primarily stemming from a thrombotic or embolic event. This impediment triggers a cascade of cellular death pathways, referred to as the ischemic cascade [194]. The optimal countermeasure is swift recanalization of the occluded artery, achievable either through enzymatic fibrinolytic treatment using recombinant tissue plasminogen activators (rtPAs) to dismantle the clot, mechanical thrombectomy to physically excise the clot, or a fusion of both modalities [195]. The prognosis of ischemic stroke patients is intimately tied to rapid recanalization success; interestingly, a deficit of recanalization results in poor outcomes in both patients with acute ischemic stroke as well as in those with atrial fibrillation [196]. However, while rtPA-mediated recanalization has been documented to enhance functional outcomes, it is not without pitfalls. Shortcomings include a modest recanalization success rate (sub-30%) and the potential hemorrhagic and neurotoxic ramifications of rtPA, especially in cases where recanalization is ineffective [197,198,199]. Additionally, the therapeutic window for rtPA is constrained, typically to 4.5 h post-stroke onset, after which its utility diminishes and risks escalate. Thus, tools to predict patient responses to rtPA could revolutionize ischemic stroke management [200].

##### Complementary Medications Enhancing Mainline Therapy

In the aftermath of rtPA infusion, the body often grapples with inadequate PAI-1 levels, impairing its capacity to regulate tPA activity and, consequently, dissolve fibrin clots. Intriguingly, while rtPA can trigger platelet aggregation and bind platelet fibrinogen without a corresponding surge in thromboxanes, adjunctive antiplatelet therapy using aspirin might counteract thrombosis post-rtPA therapy. Some research insights have highlighted that rtPA activates specific platelets that deter aggregation [201]. There’s a consensus, however, that such treatments are pivotal in thwarting extensive platelet activation subsequent to tPA therapy, and contemporary guidelines overwhelmingly support rtPA thrombolysis as a primary intervention [202].

Turning our attention to aphasia therapy, while speech and language therapy (SLT) alongside pharmacological interventions remain the cornerstone, augmentative therapeutic strategies are continuously sought to optimize the brain’s recuperative potential, especially in stroke’s chronic stages [203]. Among these, non-invasive brain stimulation (NIBS) methods, notably transcranial magnetic stimulations (TMSs) and transcranial direct current stimulations (tDCSs), hold promise. Multiple studies have documented these techniques’ efficacy in enhancing linguistic functions in aphasic patients and in fostering neuroplasticity [204] Table 1.

**Table 1 biomedicines-11-02617-t001:** Non-invasive brain stimulation in post-stroke aphasia (PSA).

Methods	Major Characteristics	Advantages	Limitations	References
rTMS (repetitive transcranial magnetic stimulation)	- Generates a magnetic pulse that influences various cerebral cortex regions. - Regularly applied pulses of a specific frequency within about 15 min. - Low-frequency stimulation (<1 Hz) inhibits cortical excitability while high-frequency stimulation (>1 Hz) amplifies it.	Balances excitability across hemispheres and realigns the linguistic network.	Requires the formulation of an optimal treatment protocol; must consider individual variability.	[205,206,207]
tDCS (transcranial direct current stimulation)	- Non-invasive and secure method for brain stimulation. - Polarizes neuronal cell membranes, modulating cortical excitation levels. - Induced cortical changes are determined by the electrode pole.	Aids in normalizing brain activity, fostering self-recovery.	Further extensive clinical trials needed for a broader PSA group.	[208]

Neuroplasticity is the brain’s inherent ability to adapt and change throughout an individual’s life. Stroke stands as a predominant contributor to long-term disability, posing not just medical but also significant economic challenges globally. While a considerable amount of research in the last decade has spotlighted neuroprotection during the acute phase of a stroke, there remains a noticeable gap in studies focusing on its chronic phase [209]. The brain’s capacity to reconfigure and restore the structure and functionality of its neurovascular networks is pivotal for recovery post-stroke. Employing a combination of adjuvant therapies alongside specific drugs might amplify the reparative processes, reinstating compromised brain functions [210]. Presently, various medications and rehabilitative strategies hold promise in promoting brain repair and enhancing clinical outcomes, even if initiated years after the occurrence of a stroke. Some pharmaceutical agents like citicoline, fluoxetine, niacin, and levodopa are either in active clinical use or undergoing trials [211,212]. Moreover, emerging research is delving into the potential of cell therapies, with our focus primarily on studies introducing cells in the stroke’s early stages. Following this, we explore pharmaceutical interventions. Herein, we’ve zeroed in on cognitive, behavioral, and physical rehabilitation techniques as well as supplementary interventions aimed at neuroprotection, like non-invasive brain stimulation and the application of extremely low-frequency electromagnetic fields. Contemporary rehabilitation strategies portray a paradigm shift towards physical interventions, considering a therapeutic window extending up to six months post-stroke. However, prior research alludes to the possibility of an even more extended window for stroke recovery [213].

Diving Deeper into Successful Pharmacological Interventions:

The identification of the ischemic penumbra in animal models and the proven efficacy of reperfusion therapies in humans once ignited optimism surrounding neuroprotection in acute stroke scenarios. However, the subsequent failures of numerous phase II and III trials led to the introduction of the STAIR recommendations, guiding both pre-clinical and clinical research [214]. Upon retrospection, it appears that the choice of agents earmarked for clinical development might not have been judicious. Despite fulfilling many STAIR criteria, the neuroprotective agent NXY-059 demonstrated no tangible benefit in a pivotal phase III study. Contemporary neuroprotective agents have yet to meet many STAIR recommendations. A myriad of unaddressed issues remain, ranging from the adoption of more representative animal models, ensuring drug distribution in human subjects, refining the evaluation metrics for neurological impairment and disability, to physiological optimization during human proof-of-concept studies. Enhancing the quality and quantity of clinical centers specializing in acute stroke research, leveraging surrogate imaging markers, and employing adaptive dose designs during phase II trials might elevate the success rates for identifying efficient neuroprotectives. Though the realm of neuroprotection in acute strokes remains fraught with challenges, its effectiveness remains a topic of debate [215]. In the face of the staggering burden that stroke presents and the limited applicability of reperfusion—currently benefiting a mere 10% of patients—there’s an undeniable call for further proof-of-concept studies concerning neuroprotection, underpinned by meticulous reviews of pre-clinical data and robust phase II trial designs [216].

Inclusive evaluations of all research articles and case series, based on their evidence level, indicated that only CPSP, excluding other central pain types, was considered. Amitriptyline and lamotrigine stood out as the sole orally administered drugs exhibiting effectiveness in CPSP management in placebo-controlled studies. Though intravenous drugs like lidocaine, propofol, and ketamine demonstrated short-term CPSP control efficacy, their potential side effects and application modes render them unsuitable for prolonged treatment. The newly introduced antiepileptic drug, gabapentin, displayed promising results in managing CPSP in select patients. It is noteworthy that amitriptyline, lamotrigine, and gabapentin collectively present a more encouraging efficacy and safety profile in contrast to traditional antiepileptic medications like carbamazepine and phenytoin, which lacked any placebo-controlled evidence advocating their efficacy. The pressing need of the hour is to conduct clinical trials that can optimize the pharmacological management of CPSP [217].

Our review encapsulated 10 distinct trials. Regrettably, most of these trials were bereft of sufficient details, making methodological quality assessments challenging. The studied drugs included piracetam, bifemalane, piribedil, bromocriptine, idebenone, and Dextran 40. Preliminary findings hint that patients undergoing treatment with piracetam demonstrated a higher likelihood of language improvement upon trial completion. Notably, there were no significant differences in the incidence of adverse effects, including death, between those administered piracetam and those on a placebo. Nevertheless, the disparity in death rates between both groups raises potential concerns regarding the elevated risk associated with piracetam intake. It remains elusive whether drug treatments offer more pronounced efficacy compared to speech and language therapy or if one drug trumps another in terms of effectiveness [218].

We embarked on a systematic review, adhering to the Cochrane methodology, that focused on randomized placebo-controlled trials examining the efficacy of antidepressants in treating or preventing depressive disorders and “abnormal mood” following a stroke. Alongside mood implications, we also analyzed the potential effects of these medications on physical and other outcomes. The available data were sourced from 7 treatment trials encompassing 615 participants and 9 prevention trials that included 479 participants. Owing to notable inconsistencies in trial design, the quality of studies, and reporting methods across the different research works, we opted against combining all the outcome data. Our findings from the treatment trials revealed that while antidepressants did mitigate mood symptoms, their impact on facilitating the remission of clinical depressive disorders was nebulous at best. Moreover, the studies did not provide concrete evidence supporting the idea that antidepressants can either ward off depression or catalyze recovery post-stroke. The current body of randomized evidence is insufficient to endorse the regular use of antidepressants for averting depression or accelerating recovery after a stroke. While these drugs might uplift mood in post-stroke patients with depression, the tangible clinical significance of such minor improvements, especially in patients without major depressive disorders, remains to be elucidated. This underscores an urgent need to further investigate the precise role of antidepressants in managing stroke aftermaths [219].

Stroke stands out as a primary contributor to severe prolonged disability among adults and ranks as the world’s second-leading cause of mortality [3]. The immediate aftermath of a stroke has historically witnessed an emphasis on rapid reperfusion and neuroprotection methodologies. By harnessing diverse mechanisms, pharmacological interventions have the potential to lessen disabilities in a significant proportion of survivors of acute strokes [220]. The inherent ability of the brain to adapt post-stroke, courtesy of its plasticity mechanisms, can be influenced by drugs [221]. The research landscape is rich with explorations into a myriad of therapeutic interventions, encompassing small molecules, growth factors, and monoclonal antibodies [222]. A noteworthy development was the discovery that the SSRI fluoxetine could ameliorate motor deficits in patients grappling with ischemic stroke-induced hemiplegia; intriguingly, this appeared to be independent of the patient’s depressive state [223]. Given these findings, it is imperative to bolster pioneering research endeavors. This will help usher in groundbreaking pharmacological treatments that prioritize neurological recuperation after a stroke, as opposed to merely focusing on acute de-occlusion and neuroprotection. Our manuscript stems from the collaborative insights of 14 seasoned scientists, and its objectives are threefold: (1) to shed light on pivotal facets of human brain plasticity post-stroke and viable pharmacological targets for rehabilitation; (2) to foster discourse on the most fitting characteristics of clinical trials assessing post-stroke recovery drugs; and (3) to set forth recommendations for upcoming clinical trials.

Several pharmacological solutions have been conceptualized for stroke treatment. They operate either by halting molecular pathways that lead to neuronal death or by bolstering neuronal survival and rejuvenation. Bar rtPA, a majority of these therapeutic agents have not made the leap from the laboratory to successful clinical trials. However, recent strides in understanding stroke’s pathophysiological nuances have reignited interest in crafting neuroprotective agents tailored for stroke intervention. This renewed interest has uncovered innovative molecular targets poised to offer tangible clinical advantages in neuroprotection and neurorestoration. Our review delves into the latest monumental advancements in stroke pharmacology, with an emphasis on ischemic stroke. We encapsulate emerging therapeutic mechanisms and cast a spotlight on the most recent clinical trial outcomes [204].

Stroke’s impact on health is undeniable, and formulating an effective treatment remains a daunting task. Bar rtPA, the number of clinically approved pharmacological solutions remains meager. Yet, the past decade has witnessed remarkable breakthroughs in unraveling molecular signaling pathways and gene expression blueprints underlying stroke’s pathophysiology. These revelations have paved the way for fresh therapeutic targets. Considerable energy has been channeled into conceiving innovative pharmacological agents and bolstering their transition from the lab to the clinic. Agents such as NMDAR antagonists, calcium channel blockers, anti-inflammatory compounds, neurotropic factors, monoclonal antibodies, and even non-coding RNAs have been rigorously probed in experimental setups. A handful, including minocycline and edaravone, have showcased their ability to elevate neurological outcomes in acute ischemic stroke patients. These successes have infused renewed vigor into translational studies centered on neuroprotectants. With an ever-growing arsenal of neuroprotective agents emerging from preclinical stages, there has been a noticeable uptick in clinical trials evaluating pharmacological strategies for acute stroke. A slew of these trials are currently in progress, signaling hope and momentum for the field [224] Figure 2.

### 4.2. Endovascular Thrombectomy

#### 4.2.1. Evolution of Endovascular Strategies over Time

Endovascular treatment (EVT) for acute ischemic stroke (AIS) has experienced significant advancements, but one of the persistent challenges remains futile recanalization. After a cerebral infarction, an inflammatory response is triggered, which plays an essential role in dictating the outcome of the recanalization procedure. This study was centered on the correlation between the systemic inflammatory response index (SIRI) and the phenomenon of futile recanalization in AIS patients [225]. By retrospectively examining patients with anterior circulation proximal arterial occlusion who underwent EVT, it was determined that factors such as age, the severity of the stroke at admission, and a higher SIRI upon admission are linked with poor functional outcomes three months post-procedure [225].

Further developments in AIS treatment techniques shed light on indications for reperfusion therapy. Beyond the already established protocol for intravenous thrombolysis using rtPA within 4.5 h and mechanical thrombectomy for large artery occlusion within 6 h, recent trials have expanded these timelines. For instance, some patients can now receive reperfusion treatment up to 24 h from the onset of their symptoms, provided they fit certain criteria and undergo necessary brain imaging [226].

#### 4.2.2. Mechanisms of Mechanical Thrombectomy

Mechanical thrombectomy has revolutionized the treatment of AIS resulting from large vessel occlusions. Over the past half-decade, numerous trials have validated its efficacy and safety for such patients, with the treatment window extending up to 24 h after the stroke onset [227]. Some landmark studies from 2015, such as MR CLEAN and ESCAPE, underscored the superiority of mechanical thrombectomy over traditional medical management in cases of anterior circulation large vessel occlusion strokes [155]. Later in 2017, studies like DAWN and DEFUSE-3 expanded the time frame for thrombectomy up to 24 h for certain patient groups [227]. Now, societal and national guidelines have incorporated these findings, recommending mechanical thrombectomy for a particular patient subset. Additionally, ongoing trials are investigating the technique’s potential benefits for other stroke subpopulations, further establishing the method’s relevance and potential [157].

Recent trials have spotlighted the combination of stent retriever thrombectomy with intravenous thrombolysis in AIS. It is now recommended for patients with large vessel occlusions when intravenous thrombolysis is not feasible, emphasizing the importance of rapid intervention post-symptom onset [228,229]. To ensure equitable access for all eligible patients to endovascular therapy, a call has been made for restructuring stroke care systems [230,231].

#### 4.2.3. Spotlight on Equipment: Delving into Stent Retrievers, Aspiration Catheters, and Their Innovations

A substantial number of ischemic strokes, approximately 10%, are categorized as large hemispheric infarcts, which can lead to severe disabilities. With the intent of minimizing the disease’s adverse effects, research has concentrated on multiple fronts, from pinpointing the predictors of malignant edema to improving surgical techniques. Decompressive surgery has emerged as a pivotal intervention, with studies delving into the nuances of managing large hemispheric ischemic strokes [232]. Neuroinflammatory responses are contingent upon the context, temporal progression, and nature of the neurological disturbance. Following ischemic stroke, brain injury manifests through necrosis and apoptosis, which subsequently instigates an inflammatory response mediated by the release of reactive oxygen species (ROS), chemokines, and cytokines. This inflammatory cascade originates in the microvasculature and encompasses various cell types, including innate immune cells like microglia and adaptive immune cells such as lymphocytes, leading to neuronal demise. Proinflammatory cytokines are implicated in numerous cerebral processes, directly impacting endothelial cells, neurons, and glial cells. Due to this intricate and sequential pathway, these cytokines might augment cellular movement or induce further damage. Post-ischemic stroke, microglia undergo a transition to the M1 phenotype, an inflammatory-promoting state that produces interleukin-1β (IL-1β), a proinflammatory cytokine with neurotoxic attributes. IL-1β can engage with the vascular endothelium, escalating leukocyte adherence and contributing to edema formation. Conversely, anti-inflammatory molecules like interleukin-10 (IL-10) and interleukin-4 (IL-4) act to negate the effects of proinflammatory cytokines. Under physiological circumstances, a nuanced equilibrium exists between proinflammatory and anti-inflammatory cytokines. However, in the context of stroke, a decline in anti-inflammatory IL-10 has been observed, correlating with improved patient outcomes, underscoring the intricate interplay between these inflammatory mediators during ischemia’s initial phases [233]. Furthermore, the function of microglia and monocytes/macrophages during cerebral ischemia is determined by their M1/M2 polarization states. The presence of specific cytokines in the surrounding environment, such as IFN-γ for M1 and TGF-β and IL-10 for M2, dictates their polarization. A predominance of the M1 phenotype is associated with exacerbated ischemic damage, activation of hypoxia-inducible factor-1 (HIF-1), and heightened anaerobic glycolysis. A shift to the M1 phenotype in microglia, coupled with an upsurge in IL-23 production, fosters the attraction and activation of γδ T cells. These represent a distinct subset of innate T cells, potentially exacerbating the detrimental consequences of acute ischemic stroke. Astrocytes, under physiological conditions, actively uptake surplus extracellular glutamate, converting it to glutamine for neuronal utilization. However, following brain injury, astrocytic damage may compromise this glutamate uptake capacity. The precise impact of ischemia on astrocytic glutamate uptake remains ambiguous, but there are indications that the expression of the glutamate transporter, EAAT2, becomes compromised during ischemic events. Two pivotal oxidative enzymes, xanthine oxidase and NADPH oxidase, are central to the synthesis of the superoxide anion, a critical radical post-stroke. In clinical observations, an independent positive correlation has been established between serum MMP-9 levels and both initial stroke severity and subsequent clinical recovery. Moreover, the permeability of the blood–brain barrier (BBB) mediated by MMP in ischemic stroke can be attenuated by cyclooxygenase 2 inhibitors. Thus, targeting Cox-2 activity or prostaglandin E2 can serve as a strategy to thwart MMP-induced BBB disruptions [233].

A growing body of evidence underscores the pivotal role of inflammation in stroke pathogenesis, marking it as a promising therapeutic target. Nevertheless, multiple studies suggest that inflammatory cells play a dual role—both beneficial and harmful—indicating that inhibiting a specific inflammatory pathway at an inappropriate time could potentially exacerbate the disease process. A more nuanced understanding of the temporal dynamics of stroke pathophysiology could inform more precise therapeutic interventions. Furthermore, integrating preclinical research using stroke models and concurrent relevant clinical conditions, such as type 2 diabetes, preceding infections, and atherosclerosis, may bridge the gap between experimental findings and clinical applications, potentially informing future successful stroke treatments.

Investigations into rodent models for PSD were categorized based on certain criteria. These models were segmented into three categories: surgical techniques, specific structures related to cognitive function, and comorbid conditions [234]. Each category emphasizes different aspects, such as the modeling technique in “surgical technique” or stroke models combined with other conditions in the “comorbid condition” category [235].

#### 4.2.4. The Benefits and Implications of Surgical Revascularization

Surgical revascularization stands out as a primary method to enhance cerebral hemodynamics, even though this particular facet has yet to be extensively scrutinized in randomized clinical trials [236]. The primary aim of revascularization is to mitigate the threat of ischemic incidents by augmenting cerebral blood flow (CBF) and reestablishing the cerebral vascular reserve (CVR). This is done through the creation of collateral pathways, stemming from observations that moyamoya arteriopathy (MMA) impacts the internal carotid arteries (ICAs) and their primary branches but leaves the external carotid arteries (ECAs) unscathed [236].

The Japan adult Moyamoya (JAM) trial stands as the inaugural prospective randomized controlled study specifically centered on the surgical treatment of MMA. Its findings highlighted the protective role of surgical revascularization against recurrent bleeding, primarily attributed to the post-operative diminishment of moyamoya vessels and the reduced hemodynamic stress in delicate collaterals [237]. The primary surgical intervention indicators encompass recurrent cerebral ischemic symptoms, cerebral hemodynamic deterioration (characterized by diminished regional CBF, vascular response, and perfusion reserve), and hemorrhaging resulting from the rupture of posterior collateral vessels [238,239,240].

For a considerable period, the surgical approach for hemorrhagic MMA was debated. However, the JAM trial has unveiled the protective effects of revascularization against subsequent bleeding, especially in patients who underwent intracranial hemorrhaging in the preceding year [239]. Delving deeper into the findings of the JAM trial, Miyamoto et al. accentuated the impact of bypass surgery on curbing the bleeding risk based on the hemorrhage site. The study illustrated a pronounced advantage of surgery for those with posterior hemorrhages in comparison to anterior hemorrhages [239,240].

In pediatric MMA cases, added vigilance is required, given the disease’s aggressive nature in younger patients. Consequently, revascularization surgeries are recommended for the majority of pediatric patients and should be executed without delay [237].

#### 4.2.5. Procedure Walkthrough: A Comprehensive Overview from Patient Selection to Post-Operative Care

A significant study comprising 333,117 patients revealed that only a minute fraction, specifically 286 (0.09%), suffered a stroke post-procedure. Since the difference between total hip arthroplasty (THA) and total knee arthroplasty (TKA) was not significant in predicting stroke occurrence, both procedures were jointly analyzed. The findings indicated that a majority, 65% of the strokes, materialized before the patients were discharged. A more detailed breakdown revealed that a quarter of these strokes took place by the first postoperative day, half by the second, and three-quarters by the ninth postoperative day. Factors like age, elevated American Society of Anesthesiologists (ASA) score, and smoking habits emerged as independent risk factors for postoperative strokes. For instance, patients in the age bracket of 60–69 had an odds ratio (OR) of 4.2, 70–79 had an OR of 8.1, and those aged 80 and above had an OR of 16.1. Similarly, patients with an ASA score equal to or greater than 3 had an OR of 1.7, while smokers had an OR of 1.6 [241].

Such findings can serve as a vital foundation for medical professionals to counsel patients about potential risks and make informed decisions to optimize patient care, ensuring the best possible outcomes after procedures like THA/TKA.

#### 4.2.6. Early Post-Operative Strokes: An Analysis of Incidence, Risk Factors, and Outcomes

In a retrospective case-control study conducted at a university-based tertiary care hospital, researchers delved into the occurrence, risk factors, and outcomes of early post-operative strokes. Specifically, they focused on cases where strokes transpired within 24 h post-surgery. By scrutinizing medical records from 2015 to 2021, they aimed to identify trends and potential preventative measures [242].

The methodology included comparing early post-operative stroke cases with age-matched controls at a 1:3 ratio. The data covering patient demographics, events during surgery, and post-operative results were meticulously analyzed. To pinpoint risk factors that could lead to post-operative strokes, a multiple logistic regression analysis was employed.

Findings indicated that early post-operative strokes, those occurring within 24 h, had an incidence rate of 0.015% (43 out of 284,105 cases). The multivariable analysis highlighted several risk factors:An American Society of Anesthesiologists (ASA) physical status of ≥3 was associated with a higher risk (adjusted OR: 3.12) [242].Surgeries lasting more than 120 min also elevated the risk (adjusted OR: 10.69) [242].Experiencing intra-operative hypotension and the usage of inotropes/vasopressors during surgery also increased the risk (adjusted OR: 2.80) [242].

When contrasted with control groups, stroke patients exhibited increased rates of both planned and unplanned ICU admissions, longer hospital stays, more frequent use of ventilators, and a higher mortality rate. Despite the relatively low occurrence of early post-operative strokes (0.015%), these incidents lead to deteriorated clinical outcomes and a surge in mortality rates. It is imperative that medical professionals recognize and address potential risk factors and strategize for optimization to diminish the incidence of these strokes [242].

#### 4.2.7. The Transformative Effects of Timely Endovascular Intervention: Key Insights from Case Studies

The American Heart Association and the European Stroke Organization have jointly endorsed endovascular thrombectomy (EVT) in specific cases. Their recommendations include using EVT, along with best medical practices, for adults experiencing anterior circulation stroke. The criteria encompass pre-stroke modified Rankin scale (mRS) scores below 2, occlusions in the internal carotid artery or middle cerebral artery, patient age over 18, and ASPECTS and NIHSS scores exceeding 6 [243,244].

Notwithstanding, there exists a debate concerning the use of EVT in patients presenting with NIHSS scores below 6. Those presenting after the 6-h window but meeting the criteria set by the DAWN and DIFFUSE 3 trials are also considered suitable candidates for EVT. The ongoing TESLA clinical trial is keen on enrolling patients with low ASPECT scores to further investigate the efficacy and safety of the procedure in this subgroup [245].

In a comprehensive review encompassing seven trials with 980 participants, distinctions were drawn between intravenous thrombolytic treatment and endovascular thrombectomy in cases of the anterior intracranial circulation’s large vessel occlusion. Notably, all these trials employed advanced imaging techniques for patient selection. The findings reflected that intravenous thrombolytic treatment resulted in a 66% good functional outcome at a 90-day follow-up, while for endovascular thrombectomy, this figure stood at 46%. However, there was a potential increased risk of symptomatic intracranial hemorrhage associated with thrombolytic treatment [246].

In conclusion, for selected patients with acute ischemic wake-up strokes, both therapeutic interventions could significantly improve functional outcomes without amplifying the risk of mortality. Yet, it remains crucial to continue researching to determine the best patient selection criteria [246].

## 5. Combined Treatment Modalities

Over the past decade, our understanding and approach to stroke management have evolved significantly. This transformation has been characterized by a broader understanding of what constitutes a stroke and notable advancements in its prevention and treatment techniques. One significant breakthrough in the treatment of acute ischaemic stroke is the synergy achieved by pairing endovascular thrombectomy for large artery occlusion with intravenous alteplase, which further enhances the likelihood of patients attaining functional independence by an additional 20%. Moreover, the protective role of aspirin in thwarting early recurrent ischaemic strokes is more potent than initially believed. The panorama of preventative strategies against recurrent strokes has expanded, now offering direct oral anticoagulants as an alternative to the traditional warfarin for atrial fibrillation patients. In addition, carotid stenting has emerged as a viable option against endarterectomy for patients displaying symptomatic carotid stenosis. Delving into acute intracerebral haemorrhage, ongoing research trials are working diligently to measure the efficacy of various interventions, from acute blood pressure regulation and haemostatic therapies to minimally invasive surgical methods, anti-inflammatory treatments, and neuroprotective techniques. A rising frontier is the use of pharmacological and stem-cell treatments to promote brain regeneration and fortify rehabilitation processes, facilitating optimal functional recovery. Studies on embryonic stem cells, mesenchymal cells, and induced pluripotent stem cells have explored their capabilities in tissue regeneration, sustenance, cell migration and proliferation, neural circuitry reconstruction, and physical and behavioral restoration. Innovations in stem cell and genomic technologies have ushered in regenerative treatments designed to reconstruct neural pathways and mend neurons affected by ischemic events. Notwithstanding the descending trend in stroke-related mortalities, the global toll of stroke remains a concern. Addressing this requires an expansive preventive approach that not only encompasses individuals across the risk spectrum but also dovetails with prevention regimes for other illnesses bearing shared risk factors [247].

For over four decades, efforts to engineer treatments that shield neurons and other brain cells from the cellular and molecular ramifications of cerebral ischaemia during acute ischaemic stroke (AIS) largely proved unfruitful. Yet, with the dawn of intravenous thrombolysis coupled with endovascular thrombectomy, the medical community has ushered in a revolutionary phase of AIS treatment. Within this realm, reperfusion therapies have emerged as potent strategies [158]. Given this new backdrop, it becomes imperative to re-evaluate cytoprotective treatments as supplementary aids to reperfusion therapy. It is essential that our energies be redirected towards designing novel drugs that holistically address multiple facets of the ischaemic cascade. Drugs developed in the past should also be revisited, especially if they showcased substantial cytoprotective effects in lab settings and were deemed safe in early human trials [248]. Multiple avenues for coupling cytoprotection with reperfusion are conceivable. This review delves deep into potential targets for cytoprotective treatment, underscoring crucial considerations for future drug formulations. A spotlight is also cast on the recent ESCAPE-NA1 trial on nerinetide, marking some of the most encouraging results yet. Furthermore, we inspect innovative clinical trials that aim to discern if cytoprotective drugs can either decelerate infarct growth preceding reperfusion or mitigate the aftereffects of reperfusion, such as haemorrhagic transformation [249].

Intracranial atherosclerosis (ICAS), characterized by its progressive nature, leads to gradual stenosis and cerebral hypoperfusion, contributing significantly to both initial and recurrent stroke incidents globally. This ailment’s emergence can be attributed to a myriad of factors. Enhanced angiographic imaging methodologies now enable more accurate ICAS diagnoses and facilitate the tailoring of appropriate therapeutic courses. Notably, neither intensive medication regimens nor endovascular interventions have managed to fully eliminate the recurrence of strokes in ICAS-affected individuals. The medical landscape is now observing the emergence of non-pharmacological therapies like remote ischemic conditioning and hypothermia [250]. A holistic therapy approach combining medication, endovascular procedures, and/or non-pharmacological treatments might be the key to the future of ICAS management. This piece provides an in-depth overview of ICAS, covering its epidemiology, underlying mechanisms, risk determinants, biomarkers, diagnostic imaging, and overall management [251].

The past few years have witnessed significant advancements in acute stroke therapy. With the introduction of thrombolytic treatments, advanced endovascular procedures, and specialized stroke care units, there has been a marked improvement in survival rates and prognostic outcomes for stroke victims. However, it is crucial to note that a significant portion of patients, especially those without access to these advanced interventions, continue to experience high stroke mortality, compounded with residual morbidity. For many, the aftermath of a stroke translates into severe motor and cognitive impairments, culminating in a loss of autonomy in daily life tasks [252]. Against this backdrop, recent research endeavors have been channeled to mitigate the cerebral damage caused by acute ischemia, termed ‘neuroprotection.’ Concurrently, there is a focus on amplifying recovery processes—encompassing plasticity, neuroregeneration, and complementing rehabilitation—to enhance recovery probabilities and facilitate a return to normal functionalities, known as ‘neurorepair.’ Citicoline, in particular, has exhibited therapeutic prowess at various junctures of the ischemic cascade during acute ischemic strokes and has showcased efficiency across diverse animal-based stroke models [253]. Chronic administration of citicoline has been proven to be both safe and efficacious, curtailing post-stroke cognitive deterioration and bolstering patients’ functional recuperation. Sustained citicoline treatment at optimal doses has been remarkably well-received, propelling intrinsic neurogenesis and neurorepair processes, which synergize with physical therapy and rehabilitation [254].

### 5.1. Global Prevalence and Management of Stroke

Stroke stands as the second primary cause of death and a principal driver of disability worldwide, witnessing a surge in incidence, particularly in developing nations [255]. A substantial portion of strokes are ischaemic in nature, arising from arterial blockages. Treatment priorities lie in immediate reperfusion using intravenous thrombolysis and endovascular thrombectomy, tools that effectively reduce disability but are immensely sensitive to time. To harness the full potential of these reperfusion therapies, it is imperative to overhaul care systems to minimize treatment delays. When administered within 4.5 h post-stroke onset, intravenous thrombolysis demonstrates a significant decrease in disability rates. Furthermore, thrombolysis has been found to aid patients showing salvageable brain tissue through perfusion imaging even up to 9 h post-stroke onset, including those who wake up exhibiting stroke symptoms. Endovascular thrombectomy has been effective in curtailing disability for patients experiencing large vessel occlusion within a 6-h window post-onset, and for those pinpointed through perfusion imaging, even 24 h after stroke onset. Prevention strategies for ischaemic stroke encompass many facets akin to cardiovascular risk management seen in other medical sectors, such as monitoring blood pressure, managing cholesterol levels, and deploying antithrombotic drugs. Additionally, certain preventive methods are tailored according to the stroke’s origin, like utilizing anticoagulation for atrial fibrillation or undergoing carotid endarterectomy for pronounced symptomatic carotid artery stenosis [256].

### 5.2. The Promise of Combined Therapies

Our objective was to evaluate the joint effects of kinesio taping (KT) and modified constraint-induced movement therapy (mCIMT) on upper limb functionality and spasticity in stroke-induced hemiplegic patients. The study, randomized and controlled, was conducted in a hospital setting on stroke patients with hemiplegia spanning 3–12 months post-stroke. A total of 35 patients participated, divided into three cohorts: sham KT with mCIMT, KT alone, or a combination of KT and mCIMT. All these treatments acted as supplementary therapies, administered alongside standard rehabilitation. The outcomes revealed significant enhancements in various scales and tests, proving that KT is beneficial for stroke patients in terms of reducing spasticity and boosting upper limb function. Importantly, coupling KT with mCIMT delivers augmented advantages in motor skills with sustained effects [257].

However, the clinical aftermath following EVT in patients suffering from proximal anterior circulation AIS can be less than ideal, even with successful recanalization. Achieving recanalization does not guarantee clinical recuperation, as reperfusion might not occur, and even when it does, it can induce damage in the brain [258]. A study pinpointed the SIRI as a determinant of unproductive endovascular reperfusion. Diving deeper into the factors and mechanisms dictating unfruitful reperfusion could pave the way for novel therapeutic avenues and strategies, potentially optimizing outcomes for acute ischemic stroke patients [225].

Moreover, frailty tends to escalate mortality and morbidity rates in patients with large vessel occlusions. With only about 20% of moderately frail patients achieving desirable outcomes, the significance of recognizing such patients and risk stratification becomes clear. The HFRS offers a means to assess these patients and provide valuable insights to their families regarding potential outcomes [259].

### 5.3. Advancements in Stroke Management

The realm of stroke management has undergone rapid evolution, solidifying its significance within the neurology discipline. Over time, there has been a refined understanding and classification of distinct stroke types, enhancing stratification and subsequently refining management strategies [260]. Therapeutically, the landscape has dramatically transformed. No longer solely reliant on IV TPA, today’s stroke management has embraced more intricate and state-of-the-art techniques. A notable stride in this domain is EVT, which, through direct clot visualization and prompt clot retrieval methods, has remarkably enhanced patient outcomes. Moreover, preventive measures against stroke recurrence have seen marked advancements, with robust evidence now supporting the use of alternative antiplatelet therapies and oral anticoagulants, further refining patient care [261].

At the heart of top-tier stroke care lies the principle of multidisciplinary involvement, starting with prehospital care and extending through post-stroke rehabilitation. This article delves into the assessment and the latest progress in managing AIS. Beginning in the prehospital setting, moving into the emergency department, and subsequently transitioning into post-acute hospital care and rehabilitation, this article captures the full spectrum of AIS management. Furthermore, it emphasizes areas in stroke care that presently show practice gaps, necessitating more comprehensive studies. A crucial note is that, in spite of the tremendous progress in stroke management, post-stroke disability remains a prevalent issue. There is an urgent need for studies focusing on refining prehospital systems for the swift recognition and transfer of stroke patients to suitable centers, ensuring prompt treatment with thrombolytics, and delineating the potential of EVT in handling posterior circulation and distal vessel blockages [262].

### 5.4. Complications, Implications, and Potential Interventions

The acute damage dealt to the brain post-stroke can spur elevated levels of catecholamine and cortisol. These heightened levels, unfortunately, can trigger the apoptosis of peripheral lymphocytes, leading to functional deactivation [263]. Given their pivotal role in cellular and humoral responses, lymphocyte loss weakens defense mechanisms against pathogens, heightening susceptibility to infections—a prevalent post-stroke complication that can exacerbate the clinical trajectory [264,265]. Notably, specific lymphocyte subgroups preserve immune homeostasis, functioning as neuroprotective agents by dampening pro-inflammatory mediators, modulating microglial activity, restraining autoreactive cells, and fostering neurogenesis and reparative processes within ischemic regions [266,267].

Our findings augment existing evidence on plasma biomarkers predicting stroke outcomes. It is vital to recognize the limitations of solely relying on single cellular line assessments, as they might not encompass the multifaceted nature of immune responses and may be susceptible to biases stemming from conditions such as overhydration, dehydration, and blood specimen handling [268,269]. A noteworthy indicator is the SIRI, which not only correlates directly with the risk of unfruitful recanalization but also complements existing clinical predictors like age and baseline stroke severity. It encapsulates the equilibrium between innate and adaptive immunity, with elevated values hinting at heightened innate immune activity and diminished adaptive immunity. In the acute stroke phase, SIRI can symbolize the immediate inflammatory response to brain damage, shedding light on the likelihood of secondary impairments and susceptibility to ensuing complications [225].

Lastly, it is imperative to mention ischemic brainstem strokes, which account for 10% of all ischemic cerebral strokes. The onset of hemorrhagic complications spells a particularly grim prognosis. Symptoms span vertigo, cranial nerve manifestations, and both crossed and uncrossed corticospinal tract findings. In the decision-making process for brainstem stroke management, avant-garde neuroimaging techniques have emerged as indispensable. They hold potential for pinpointing patients who might benefit from thrombolysis. However, given the paucity of substantial data guiding many current recommendations, there is a palpable need for further research dedicated to acute brainstem stroke treatment [270].

### 5.5. Immune Dynamics Post-Stroke

The aftermath of a stroke reveals that the brain’s damage is not just a simple mechanical disruption but rather intricately interlinked with immune processes. When the brain gets assaulted during a stroke, it spirals into an inflammatory state, eliciting a chain reaction involving the sympathetic nervous system and the hypothalamic-pituitary-adrenal axis. Consequently, this leads to the production of immune-suppressing compounds such as catecholamines and glucocorticoids [271].

Newer research has shed light on the brain-immunity axis, a sophisticated pathway responsible for inducing immune suppression following a stroke. The common belief is that this suppression is the brain’s proactive measure to curtail additional cerebral damage. But this defensive stance comes at a cost: it leaves the patient more vulnerable to infections. Conditions like pneumonia and UTIs become frequent adversaries for stroke patients, often leading to aggravated health outcomes and a heightened mortality rate [272].

Given these findings, the medical fraternity has pivoted its focus toward minimizing the risk of post-stroke infections. The idea of preventive antibiotic treatment initially garnered interest; however, a deeper dive into clinical studies indicated that such prophylactic measures did not necessarily bring down pneumonia rates or enhance clinical results. One significant concern stems from discrepancies in pneumonia diagnosis standards across healthcare facilities. With the looming shadow of antibiotic-resistant strains, the community is nudged to explore other avenues. Immunomodulation therapies, which bolster the immune system’s ability to tackle infections, appear promising. Yet, one must tread cautiously to ensure that these interventions do not inadvertently intensify cerebral damage. As we move forward, it is imperative for research to delve deeper into the precise dynamics of immune suppression post-stroke, potentially revealing novel therapeutic avenues to counteract associated infections [273,274].

## 6. Envisioning Comprehensive Patient Care

*Merging Physical Rehabilitation with Acute Stroke Care*:

Post-stroke therapies bear a dual focus: while neuroprotection aims to minimize immediate stroke-induced damage, repair therapies target cerebral restoration. The therapeutic landscape is rife with potential innovations. For instance, “mirror therapy”, seamlessly blending with conventional physical therapy approaches, emerges as an easily applicable and effective technique [275]. On a similar note, neuromuscular electrical stimulation has been lauded for its ability to rejuvenate neuromuscular function and kickstart cerebral adaptability [276].

When it comes to augmenting motor function, combining transcranial magnetic stimulation with standard physical and occupational therapies has proven fruitful. The enhancement arises from a heightened stimulation of the motor cortex regions on the side opposite the hemiplegia [277]. Of notable mention is the ongoing NEST-3 trial, a rigorous exploration of the NeuroThera^®^ Laser System’s efficacy in treating acute ischemic stroke patients within the crucial 24-h window post-ictus [277]. Additionally, the horizon of stroke therapy seems bright, with robotic interventions demonstrating immense potential.

2.*Pharmacological Enhancements Post-Stroke*:

Pharmacotherapy holds promise in the post-stroke recovery phase. Specifically, a range of anti-depressants, including serotonin uptake inhibitors (SSRIs) and noradrenergic inhibitors, have shown potential for amplifying motor recovery in ischemic stroke victims [278,279]. However, the inner workings of how SSRIs achieve this remain elusive and warrant further exploration [254] Table 2.

### 6.1. Objective Evaluation of Non-Pharmacological Therapeutic Interventions in Acute Ischemic Stroke

In assessing the efficiency of contemporary non-invasive, non-pharmacological therapeutic and rehabilitative interventions for acute ischemic stroke, it is crucial to employ objective, quantifiable measures. Integrating these metrics can make a significant difference in clinical decision-making. A myriad of tools exist for these evaluations:Clinical-functional evaluation tools: These instruments, like the Glasgow coma scale (GCS), Glasgow outcome score scale (GOS), modified Rankin scale, and the national institutes of health stroke scale (NIHSS), provide key insights into a patient’s neuro-functional state [283].Neuroimaging and Neurophysiological Examinations: Technologies such as structural and functional magnetic resonance imaging, positron emission tomography (PET), single-photon emission computed tomography (SPECT), and repetitive/transcranial magnetic stimulation (r/TMS) give in-depth visuals and data about the brain’s condition. Additionally, considering a patient’s general health and neuro-functional state, tools like functional near-infrared spectroscopy (f/NIR) and diverse immuno-(cyto)/histochemical assays could also be invaluable. Similarly, pairing transcranial magnetic stimulation (TMS) with high-density EEG offers a non-invasive method to perturb and measure, providing insights into both local neuronal conditions and signal dispersion within functional networks. Notably, even in patients who appeared clinically identical (e.g., lacking residual arm functionality or lacking peripheral motor-evoked potential from standard TMS), TMS-EEG unveiled varied response patterns that correlated with subsequent recovery. This underscores the profound potential of TMS-EEG as a novel indicator of the motor network’s functional reserve [284].

### 6.2. Cardiac Connections to Stroke

Ischemic strokes often coexist with cardiac abnormalities, like myocardial ischemia. This connection is evident in the ECG manifestations, such as depressed ST-segments, prolonged QT-interval, and alterations in T and U waves. A stroke can perturb the autonomic function, resulting in heightened sympathetic activity. Thus, variations in ECG heart rate variability serve as clinical markers, illuminating alterations in the autonomic nervous system (ANS) post-stroke [285]. Non-invasive assessment tools like electrocardiography (ECG), electroencephalography (EEG), and electromyography (EMG) offer predictive insights into stroke prognosis [286,287]. The evolution of technology has enabled machine learning to analyze vast ECG datasets, providing preliminary stroke prognosis and tracking post-stroke recovery using cardiac activity profiles [288].

### 6.3. Rehabilitation after Stroke: An Ongoing Journey

The systematic practice of daily activities has been identified as a potent strategy to boost mobility and activities of daily living (ADLs) in stroke survivors [289]. A crucial aspect of post-stroke care is the rehabilitation process. This proactive journey commences soon after hospitalization, transitioning through structured rehabilitation regimens and continuing even after the patient reintegrates into the community. Typically, a stroke patient starts low-intensity rehabilitation as early as 72 h after the event, often in specialized units or ICUs. Along with the patient, caregivers receive guidance on the path of recovery—its duration, the care trajectory, foreseeable limitations, and more—all navigated with the assistance of the healthcare team.

A study from 2002 investigated various rehabilitation strategies for acute stroke patients. Participants were divided into three intervention groups: standard care, functional task practice, and strength training. The results emphasized that task specificity and stroke severity are key considerations for post-stroke rehabilitation of the upper extremity. Significantly improved functional outcomes were achieved with targeted upper extremity treatment spanning four to six weeks. While both functional task practice and resistance strength training offered immediate advantages, the former showcased long-term benefits [290].

For acute stroke rehabilitation, interventions like early mobilization, positioning, functional mobility training, ADLs training, range of movements (ROMs), splinting, and bed mobility are vital. Mirror therapy stands out, positively influencing motor deficits, emotional wellness, visuospatial neglect, and post-stroke discomfort. As rehabilitation progresses to the sub-acute phase, the focus shifts to enhancing mobility, stamina, strength, and equilibrium. Past research has shown aerobic exercises to be beneficial even beyond the sub-acute stage [291]. Additionally, early overground bodyweight-support training has demonstrated advantages in the sub-acute phase [292]. As patients transition to the chronic stage, task-specific therapies offer persistent improvements in diverse motor deficits [293].

### 6.4. Psychological and Cognitive Repercussions of Stroke

Post-stroke cognitive impairment and dementia (PSCID) is an increasingly recognized aftermath of stroke events globally, often leading to significant morbidity and mortality. This impairment can be a consequence of various forms of stroke: ischemic, intracerebral hemorrhage, or subarachnoid hemorrhage. Even if a patient has underlying neurodegenerative pathologies—a common scenario in the elderly—cognitive decline after a stroke is typically attributed to vascular causes. Moreover, cerebral small vessel disease manifestations such as covert brain infarcts, white matter disruptions, microbleeds, and cortical microinfarcts are frequently observed in these patients and contribute further to cognitive decline.

Several factors escalate the risk of PSCID. Among them are age, lower educational attainment, socioeconomic challenges, pre-stroke cognitive or functional decline, lifelong vascular risk exposure, and prior stroke history. Additionally, the specifics of the stroke event, such as its severity, lesion volume and location, and recurrence, play a role in determining PSCID risk.

A comprehensive understanding of how acute stroke events and pre-existing brain pathologies interplay is crucial. It holds the promise of refining personalized predictions, preventive measures, interventions, and rehabilitation strategies tailored to individual needs. While the field has made progress, it still grapples with standardizing definitions, diagnostic timing, neurocognitive assessment methods, and follow-up durations post-stroke. Nevertheless, emerging insights from pathophysiology studies, advances in neuroimaging, and biomarker discoveries offer hope for clinical innovations and prospective trial designs. A pivotal strategy for optimal brain health hinges on preventing both strokes and the resultant PSCID [294].

Stroke, predominantly observed in the elderly, has profound implications for cognitive function. The repercussions, manifesting as post-stroke cognitive impairment (PSCI), not only challenge the affected individual but also place significant burdens on caregivers and society at large. The gravity of PSCI necessitates efficient management, particularly preventive approaches that address modifiable risk factors. In shedding light on PSCI, we explore its varied definitions and inconsistent prevalence findings across studies, delve into established and potential predictors, and discuss prevention and treatment avenues currently in use or under trial. The goal is to direct future research towards bridging existing knowledge gaps [295].

Ischemic strokes bear a global burden that is nearly four times greater than hemorrhagic strokes. Recent data indicates that between 25–30% of ischemic stroke survivors experience immediate or subsequent vascular cognitive impairment (VCI) or vascular dementia (VaD). Dementia, post-stroke, can cover a spectrum of cognitive disorders. Instances of cognitive dysfunction present before the actual stroke fall under pre-stroke dementia. This condition could result from vascular alterations or gradually advancing neurodegenerative processes.

Several risk factors for post-stroke cognitive impairment have been identified: aging, genetic predispositions, educational background, vascular-related comorbidities, prior transient ischemic attacks, recurrent strokes, and associated depression. Neuroimaging has revealed that silent brain infarcts, alterations in white matter, lacunar infarcts, and atrophy in the medial temporal lobe are indicators of post-stroke dementia. Historically, the neuropathological understanding of post-stroke dementia was limited. Now, it is largely attributed to VaD, with a blend of multiple underlying factors. Key contributors include microinfarction, changes in the microvascular system linked to blood–brain barrier damage, localized neuronal atrophy, and a minimal presence of co-existing neurodegenerative pathology. To mitigate the cognitive challenges post-stroke, understanding the underlying mechanisms is vital. Effective strategies for alleviation and prevention hinge upon rigorous control of vascular disease risk factors [296].

### 6.5. Education and Intervention: Addressing the Rising Incidence of Stroke

In recent decades, there has been a concerning rise in the incidence of strokes, notably ischemic strokes, among young adults. This surge parallels increasing rates of traditional risk factors such as hypertension, diabetes, and tobacco use, along with the consumption of illicit substances. Young adults present a unique challenge as they exhibit a higher proportion of intracerebral and subarachnoid hemorrhage when compared to older individuals. Moreover, the root causes of ischemic strokes in young adults differ significantly. One-third of these strokes have undetermined etiologies, often due to insufficient investigation into the multitude of potential causes. It is alarming to note that young individuals with premature atherosclerotic cardiovascular disease receive inadequate secondary prevention care, marked by lower utilization of antiplatelets and statin therapy. Factors such as atypical stroke symptoms, varied causes, reluctance to use statins, and clinical inertia on the part of the provider pose barriers to timely diagnosis and treatment. To mitigate the long-term implications of stroke in this demographic, immediate recognition, rigorous risk modification, and an emphasis on both primary and secondary prevention therapies are paramount [297].

Early risk factor modification holds significant promise for preempting strokes. Elevated awareness and intervention, especially regarding hypertension, can drastically diminish stroke-related morbidity and mortality. New criteria have expanded the pool of individuals classified as hypertensive, thereby widening the reach of lifestyle modifications and medical interventions. Direct oral anticoagulants, which have simplified the treatment of patients with atrial fibrillation, are now recommended as primary therapy for those with added stroke risks. In addition, it was observed that the risk factors for acute ischemic stroke are the same as those for atrial fibrillation; therefore, the underlying mechanisms of both pathologies are triggered by the same causes. The bedrock of preserving brain health through one’s lifespan is the rigorous primary prevention of stroke, emphasizing a wholesome lifestyle and routine screening for predisposing factors [298].

In China, strokes account for the highest number of disability-adjusted life years lost among all diseases, with over 2 million new cases diagnosed annually. This number is projected to grow, propelled by an aging populace, persistent high-risk factors like hypertension, and subpar management. While there is enhanced access to general healthcare, specialized stroke care remains disparate, with rural areas being disproportionately disadvantaged. Hospital outcomes are better today due to the availability of reperfusion therapies and supportive care. However, there is a noticeable shortfall in adherence to secondary preventive measures and sustained care. Even though global standards of care, such as thrombolysis and dedicated stroke units, are acknowledged in China, the uptake is slow due to concerns about bleeding risks and logistical hurdles. Herbal treatments and neuroprotective drugs, although not strongly evidence-backed, are prevalent in the Chinese context. The prolific use of advanced neuroimaging has also inadvertently led to the overdiagnosis and overtreatment of ‘silent strokes.’ China’s focus must shift towards ensuring equal access to specialized stroke care, augmenting evidence-based practices, and fostering translational research to ameliorate outcomes for stroke patients [299].

Globally, despite leaps in early diagnosis and aggressive vascular risk factor treatments, stroke retains its grim position as a principal cause of death and long-term disability. It is essential to recognize the disparity in stroke risks, occurrences, and treatments. Stroke’s multifaceted nature, influenced by a myriad of additive risk factors, necessitates a tailored approach to both primary and secondary prevention, with the former targeting risk mitigation and the latter emphasizing treatments tailored to the initial stroke or transient ischemic attack’s cause [300].

Addressing the escalating global disability attributed to strokes and the concurrent rise in stroke care costs necessitates a research pivot towards efficacious stroke prevention measures. Strategies span from reducing disease emergence risks (primordial prevention) and forestalling disease onset (primary prevention) to averting disease recurrence (secondary prevention) [301]. A comprehensive approach requires worldwide collaboration involving global strategies, campaigns, and effectiveness measurement of global preventive initiatives, especially in low- to middle-income countries. Findings underscore the importance of tobacco control, nutritional adequacy, and fostering healthier urban environments for primordial prevention. In contrast, primary prevention can leverage polypill approaches, mobile health technologies, and dietary interventions like salt reduction. Effective intersectoral collaboration, steered by robust government policies and campaigns, can seamlessly implement secondary prevention strategies. Tools like the WHO’s non-communicable disease programs, spread across both affluent and developing nations, further underscore the promise of such collaborations [302].

### 6.6. Community Engagement and Support: Building Resilience among Stroke Survivors

A comprehensive review of current literature was undertaken by searching leading databases, including MEDLINE, Cochrane Central, and Web of Science. Utilizing, the extracted continuous data from randomized controlled trials (RCTs) was meticulously analyzed to discern any potential variance in outcomes [303]. Out of 15 studies involving 1339 patients that were reviewed, a mere 12 were deemed suitable for the pooled analysis. These examinations compared the results between groups undergoing telerehabilitation and those without. Intriguingly, outcomes measured through the Barthel index, Berg balance scale, Fugl–Meyer upper extremity, and the mobility subscale of the stroke impact scale presented no significant differences between the groups [304]. Furthermore, when assessing the health-related quality of life of stroke survivors, the strain on caregivers, and the overall satisfaction with the treatment provided, both groups appeared equally matched [305]. It is worth noting that telerehabilitation presents a viable alternative, especially for patients in remote or underserved locations. Nevertheless, future studies with larger sample sizes are encouraged to further delve into the intricacies of the health-related quality of life and economic implications of enhanced telerehabilitation networks [306].

Shifting our focus to home-based rehabilitation (HBR), an innovative study aimed to harness the capabilities of machine learning (ML) to track and recognize rehabilitation exercises using a smartwatch and smartphone app. This study showcased an ingenious system that integrated a commercial smartwatch with specially designed apps, using a convolutional neural network to detect home exercises [307]. An evaluation period from March 2018 to February 2019 saw two groups of stroke survivors being assessed. Remarkably, the system achieved almost near-perfect accuracy, particularly when using both accelerometer and gyroscope data. Furthermore, the study found that the HBR group exhibited significant improvements in specific clinical tests compared to the control group. The promising results suggest that such a system could be a game-changer for cost-effective, home-based care for stroke survivors.

On a global scale, strokes remain a formidable challenge, affecting over 80 million individuals and exerting substantial economic and societal costs [308]. Despite commendable strides in acute care, there remains a palpable lack of focus on post-acute stroke care. Recent evaluations have highlighted the existing gaps and challenges in post-acute care, urging a paradigm shift to encompass this critical area of stroke management [309]. Three bold recommendations were put forth: first, the establishment of criteria for rehabilitation readiness within Comprehensive Stroke Centers. Second, an expansion of the American Heart Association/American Stroke Association’s guidelines to cover rehabilitation readiness and a 90-day outcome measurement. Lastly, a targeted public health campaign to amplify the message of secondary prevention and recovery. These suggestions underscore the pressing need for an all-encompassing, holistic approach to stroke care, valuing not just acute interventions but also long-term recovery and support.

## 7. Future Horizons in Stroke Management

### 7.1. On the Brink of Discovery: The Future of Diagnostic Techniques and Treatments

In the aftermath of an acute ischemic stroke (AIS), the body’s inflammatory reactions can be a double-edged sword, having both potential advantages and disadvantages. Recent evidence sheds light on the inflammatory pathways and mediators that are currently being explored as potential therapeutic targets [310]. A comprehensive search was conducted on platforms like PubMed and MEDLINE until August 2016, using keywords that range from ischemic stroke to autoimmunity. The results highlighted the surge of cytokines, chemokines, and damage-associated molecular patterns (DAMPs) in AIS, which can exacerbate tissue damage during both the immediate response and the repair phase. Despite the plethora of biomarkers examined, none fully captured the intricate nature of the systemic immune response. Even though reperfusion therapies have shown promise in AIS recovery, truly effective long-term anti-inflammatory interventions remain elusive. The horizon does, however, look promising with advances in therapies such as monoclonal antibodies and cell-based treatments [310].

The burdens of IS (ischemic stroke) remain pressing, even with clinical advancements made over the years. In the context of inflammation following an IS, there is a distinction between central and peripheral responses based on origin—either the brain or peripheral tissue. Macrophages, particularly the differing responses of M1 and M2 types, have been spotlighted in recent studies post-IS. Interestingly, the spleen has been identified as a potential hotspot for inflammatory cells and cytokines after IS [311]. Beyond the commonly recognized cytokines and chemokines, new players like OPG, OPN, and autoantibodies are emerging as crucial mediators in the inflammatory environment post-IS [312]. As the medical world embraces newer treatments like monoclonal antibodies and fingolimod, it is clear that there is still much ground to cover. Some treatments have shown promise in initial clinical trial phases, while others, like cell-based therapies, could even extend their benefits beyond stroke to other neurological conditions [310].

Robotics in neurointervention is a rapidly advancing frontier. Whether it is diagnostic angiography or robot-assisted aneurysm coiling, success has been observed [313]. The rapid evolution of internet connectivity, particularly with the advent of 5G, will further catalyze the deployment of robotic systems. Such systems, while presently being operated from within the catheterization labs, have also shown promise in remote operations. A groundbreaking trial saw a telerobotic-assisted percutaneous coronary intervention being performed with the operator situated 20 miles away [314,315].

A significant stride forward in stroke prediction is the development of an ML (machine learning) pipeline, framing the prediction as a binary classification challenge. By employing a range of ML models, the study managed to achieve impressive results, particularly a low false-negative rate of 18.6% [316,317]. A significant achievement was the identification of pivotal risk factors contributing to the prediction of stroke occurrence. This study used a robust methodology, combining data resampling to address data imbalance issues with SHAP-based explainability analysis. By enhancing the understanding of stroke mechanisms through cutting-edge ML methods, this research paves the way for the development of potent diagnostic tools that can provide clinicians with a more reliable and non-invasive prediction capability [317].

### 7.2. Advancements in Stroke Prediction: The Role of Machine Learning and Biomarkers

Identifying reliable biomarkers and parameters that predict functional outcomes, as measured by mRS upon discharge, is crucial. To achieve this, a comprehensive ML pipeline was set in motion. In the foundational stage, data underwent normalization, wherein feature standardization occurred by eliminating the mean and scaling to unit variance. This ensured that subsequent feature selection and learning phases had a consistent reference point [318]. The Boruta library, which builds upon the principles of a random forest classifier, was utilized to select and rank features based on their importance.

The study incorporated five renowned classifiers: random forest (RF), XGBoost, multi-layer perceptron (MLP), support vector machines (SVMs), and logistic regression (LR). Their inclusion was driven by a need to gauge their suitability for the task at hand. Hyperparameter tuning was integral to preventing overfitting and optimizing the efficacy of the classifiers. RF and XGBoost, both ensemble learning algorithms, were chosen owing to their commendable speed and performance [319]. In contrast, MLP, which mirrors the intricate design of human neural networks, can adeptly handle multifaceted data. Additionally, the study leveraged SVMs, which excel in high-dimensional data scenarios, and the LR model, recognized for its prowess in binary classification tasks [320].

The implementation process consisted of two distinct steps. The initial step utilized random forest regression to fill in data gaps before classification. Following this, an automated hyperparameter optimization (AutoHPO) rooted in deep neural network (DNN) methodologies was applied to predict stroke in a dataset where occurrences were unevenly distributed. Of the 43,400 medical records examined, 783 instances of stroke were identified. Notably, the prediction method reported a false negative rate of just 19.1%, marking a significant 51.5% improvement over conventional techniques. Additionally, false positive rate, accuracy, and sensitivity stood at 33.1%, 71.6%, and 67.4%, respectively [316].

### 7.3. The Revolutionary Impact of AI and Machine Learning in Stroke Management

In the realm of stroke diagnostics and prognostics, tools enhancing diagnosis speed and precision can be life-changing. This is where artificial intelligence and machine learning (AI/ML) promise transformative advancements. Healthcare adoption of AI/ML, currently growing at a rate of 40% annually, has the potential to yield a staggering USD 150 billion in savings by 2026 [321]. Recognizing AI/ML’s revolutionary capabilities, the U.S. Food and Drug Administration (FDA) has inaugurated fresh protocols to evaluate AI/ML-empowered health tools in terms of safety and effectiveness. Various clinical avenues, ranging from liver fibrosis detection to lung cancer classification, have started harnessing the power of AI/ML algorithms [322,323,324].

Of significance, the FDA has green-lighted 22 AI/ML technologies designed specifically for stroke diagnosis and rehabilitation. While previous literature reviews have predominantly centered on AI/ML algorithms curated for research, there is an absence of comprehensive studies on the real-world clinical efficacy of FDA-endorsed devices tailored for stroke diagnosis and management. This review seeks to consolidate the most pertinent and current insights related to these technologies, offering an exhaustive overview of their distinctive features and contributions to refining clinical practices and outcomes. In essence, a total of 22 innovative AI/ML technologies have received FDA approval, aiming to support clinicians in diagnosing or managing strokes, or ICH [325,326,327].

### 7.4. The Technological Revolution in Stroke Diagnostics and Rehabilitation

The diagnosis and treatment of stroke have experienced unprecedented advancements with the introduction of 20 groundbreaking technologies. By prioritizing and triaging crucial scans, these innovations not only mitigate the necessity for labor-intensive tasks like segmentation but also expedite the intervention process, thereby offering patients quicker access to life-saving treatments [328]. Testament to their efficacy and acceptance, technologies like RAPID and Viz.ai have been adopted by 1800 and 900 hospitals, respectively.

Interestingly, many of these technologies seamlessly integrate into broader technological frameworks to enhance care coordination. For instance, solutions offered by companies such as Viz.ai and RapidAI are housed within an interconnected ecosystem that encompasses mobile notifications, remote CT/MRI viewing capacities, and ensures HIPAA-compliant peer communication. This holistic approach not only streamlines care coordination but also amplifies the benefits of AI/ML, creating a harmonious synergy that has substantially improved patient care outcomes. The trajectory suggests that future technologies should lean towards such integrated clinical systems. Historical data shows that technological integration and workflow simplification have notably improved outcomes, even in non-stroke medical emergencies.

Moving beyond diagnosis, AI/ML is also redefining post-stroke rehabilitation. The two current devices focused on rehabilitation harness AI/ML to innovate therapeutic techniques. These have ushered in newfound hope for patients, enabling substantial recovery after severe neurological damage. As these devices mature, they promise even greater safety and efficiency, equipping healthcare professionals with advanced tools to ensure superior patient recovery. Yet, the journey forward requires rigorous head-to-head comparisons using identical clinical datasets to guide clinicians in their choice of technology [329].

In the domain of medical imaging, the present algorithm leverages binary masks of ICA and CCA images for model training. This process mandates meticulous mask preparation, which, if overlooked, can lead to model misrecognition. However, this task is painstakingly slow and requires seasoned sonographers [330]. With the data influx, generating masks for thousands of images becomes impractical. Consequently, supervised learning faces challenges in massive data analyses [331], suggesting that unsupervised models might soon become the standard, especially in contexts where binary masks or labeled data are not mandatory. Medical image processing has vast untapped potential for such unsupervised deep learning models [332].

### 7.5. Deep Dive into EEG Data for Stroke Prediction

Our research ventured into deciphering stroke predictions via ML models, analyzing EEG data from both stroke-afflicted and healthy individuals during active states like walking, working, and reading. Distinct EEG band power is indicative of brain functionality and, in ischemic stroke contexts, correlates with neural damage in the brain’s lesioned region. Employing cutting-edge ML models like adaptive gradient boosting, XGBoost, and LightGBM, our study aimed to categorize active-state stroke patients. Although XGBoost and LightGBM demonstrated commendable performance metrics, the adaptive gradient boosting model surpassed them across all parameters. Its superior precision and recall metrics underscore its proficiency in distinguishing stroke patients from non-patients with heightened accuracy [286].

Further, our research delved into explainable ML methodologies, focusing on the automated diagnosis of ischemic stroke victims and healthy individuals using neural markers from active states [333]. Notably, both the Eli5 and LIME interpretable models assign paramount importance to delta and theta waves during stroke patient identification. By making the diagnostic process transparent, the results from our exploration of explainable AI have the potential to revolutionize stroke treatment and rehabilitation, in turn facilitating physicians in elucidating their diagnostic rationale [334].

### 7.6. The Evolution and Potential Future of Cerebrovascular Disease Management

The journey of cerebrovascular disease management has been marked by considerable advancements. Yet stroke remains a predominant cause of mortality and debilitation on a global scale. Over the past decade, the stride towards conquering this ailment has been particularly notable, especially with the inception of novel orally active anticoagulant drugs [335].

A review of these modern milestones reveals an intricate interplay of anticoagulants and antiplatelet agents in tailoring the approach to distinct ischemic stroke subtypes. These agents have been a beacon of hope in both the prevention and treatment spheres. The current clinical guidelines have a tripartite recommendation system: (a) During the early stages of an ischemic stroke, aspirin—an antiplatelet agent—is favored. (b) For recurrent prevention of noncardioembolic ischemic strokes, primarily attributed to large artery atherosclerosis, antiplatelet therapies like aspirin, clopidogrel, and dipyridamole are advocated. (c) In cases of cardioembolic ischemic strokes, particularly those linked to atrial fibrillation, anticoagulants, encompassing warfarin and the new-age NOACs, are suggested [336].

Phase III clinical trials have spotlighted NOACs like dabigatran (150 mg BID) and apixaban (5 mg BID) as being superior to the traditional warfarin in staving off stroke and systemic embolism. Rivaroxaban (20 mg QD), meanwhile, showcased equivalence in efficacy. A parallel between the bleeding tendencies of dabigatran, rivaroxaban, and warfarin was observed. However, apixaban’s reduced major bleeding incidents make it stand out. As the utilization of these avant-garde anticoagulants widens, in tandem with diverse clinical investigations, it is anticipated that their efficacy in decreasing stroke instances and the complications tied to warfarin will be further emphasized.

Pivoting to the management of acute ischemic stroke, we see a domain in flux, primarily driven by the impressive outcomes of mechanical thrombectomy. Large vessel occlusion (LVO), accounting for approximately 38% of acute ischemic strokes, once brought profound distress to patients and their kin in the pre-intervention era [337]. The advent of mechanical thrombectomy heralded a revolutionary shift in LVO management, backed by a series of randomized controlled trials across nations that underscored its immense benefits. This article aims to present an exhaustive overview of LVO handling, the techniques and apparatuses employed, and potential future avenues in stroke treatment.

Historically, the mid-1990s saw the discovery of the intravenous tissue plasminogen activator (tPA), offering a modicum of benefit [338]. But its constraints—a narrow therapeutic window, minimal efficacy in vessel recanalization, and favorable outcomes—limited its universal adoption. The goal has always been to achieve results paralleling the cardiac medical domain. This aspiration became more tangible in 2015, when mechanical thrombectomy for acute ischemic strokes was rejuvenated, following the successful conclusions of five crucial RCTs [339]. These trials, spanning multiple countries, unanimously echoed the profound advantages of mechanical thrombectomy, charting a promising trajectory for cerebrovascular disease management.

### 7.7. Bridging Research and Clinical Practices in Stroke Rehabilitation: A Comprehensive Exploration

Stroke’s prevalence and associated disability rate warrant an urgent need for effective rehabilitation methods. Rehabilitation serves as an invaluable tool for curbing the disabilities that arise from strokes. Central to crafting impactful rehabilitation strategies is the art of systematic assessment. Here, we delve into prevalent methods employed both in research and clinical settings, which range from specialized stroke rehabilitation scales to sophisticated biomedical technologies.

These technologies, from surface electromyography (sEMG) and motion analysis systems to transcranial magnetic stimulation (TMS) and magnetic resonance imaging (MRI), offer diverse avenues for understanding and assessing the stroke’s impact and the progress of rehabilitation. Furthermore, innovative combinations of these techniques are charting new frontiers in assessment strategies. On the horizon, we observe pioneering experimental techniques such as the integration of artificial intelligence (AI) with optical correlation tomography (OCT) that are poised to redefine stroke rehabilitation assessment [340].

Yet, while traditional assessment scales are a staple in clinical scenarios, they often grapple with challenges related to consistency, stability, and objectivity. Biomedical techniques, despite their promise of objective data, are often stymied by the absence of comprehensive clinical studies and cohesive usage guidelines [341]. Looking forward, two trajectories are emerging in assessment development. First, there is a push towards harnessing novel technologies to deepen our understanding of post-stroke brain recovery. Second, there is momentum behind leveraging AI algorithms to replace older, manual methods, aiming for enhanced objectivity, precision, and standardization. But these pioneering technologies, still in nascent stages, need considerable research and validation before mainstream clinical adoption [342].

Shifting focus to animal models, we understand that perfectly replicating human stroke in animals is an elusive goal. The complexity and heterogeneity of ischemic stroke in humans render this task intricate. The middle cerebral artery occlusion (MCAo) model stands out due to its ability to closely mirror human ischemic strokes and its consistency in producing reproducible infarcts. While the MCAo model dominates research studies, alternative models using thromboembolic clots also hold merit, especially when evaluating thrombolytic agents. However, the current state of preclinical stroke research suffers from a low rate of translation into successful clinical outcomes. Factors such as the model choice and discrepancies between therapeutic windows in animal models versus humans can be attributive. Moreover, the prevalent use of young animals without comorbidities for these studies creates a disconnect from human stroke scenarios where elderly individuals with multiple cerebrovascular risk factors are most affected. Thus, the need for optimizing animal models to closely represent human conditions is imperative [343].

On the genetic front, strokes do not merely arise from environmental factors; they also have a hereditary component. The ever-evolving field of genetics has pinpointed numerous single-gene disorders linked to strokes and identified common variants across roughly 35 genetic loci correlated with stroke risk. Such findings have been instrumental in unveiling new stroke mechanisms and shared genetic influences across vascular conditions. The power of genetics has further illuminated causal relationships with risk factors and is pivotal in prioritizing clinical trial targets. An upcoming challenge lies in harnessing genome-wide polygenic scores to identify at-risk individuals even before any vascular risk factors manifest. Moving forward, understanding the nuances of rare genetic variants, appreciating ancestral genetic differences, integrating genetics into precision medicine, amalgamating omics data, and translating these genetic discoveries into transformative therapies will be critical [344].

## 8. Conclusions

### 8.1. Interconnected Aspects of Stroke Management: A Comprehensive Overview

The multifaceted nature of stroke management highlights the urgent need for timely, cohesive, and informed medical intervention. Acute ischemic stroke, by its very nature, places a premium on time. Ensuring optimal outcomes necessitates a seamless collaboration spanning the gamut of medical professionals, from emergency care providers to neurologists and interventional neurologists. Rapid and accurate diagnosis, expedient patient stabilization, and judicious imaging choices are cornerstones of effective care. Any delay or misstep in these processes may lead to diminished functional outcomes and even permanent paralysis [15]. As medical advancements continue to evolve, specialized stroke response teams are emerging, akin to cardiology teams that address acute myocardial infarction. However, challenges such as public awareness and hospital preparedness loom large, necessitating dedicated efforts to surmount them.

Amidst this backdrop, the past decades have seen transformative innovations heralding a renewed era in vascular neurology. This has expanded the treatment bracket and subsequently improved patient outcomes. Yet troubling trends are emerging, particularly in the US, where, despite advancements, stroke mortality rates in several states appear stagnant or even regressing. A proliferation of stroke risk factors such as diabetes, hypertension, and hyperlipidemia underscores this concerning trend [345]. To pivot this trajectory, there’s a growing consensus on the pivotal role of patient education and prevention in mitigating the incidence of debilitating or fatal strokes.

Technological advancements, notably the integration of AI/ML in clinical tools, are propelling stroke care into a futuristic paradigm. As of now, the FDA has sanctioned 22 distinct AI/ML-empowered technologies tailored to aid clinicians in stroke diagnosis or management. A majority of these tools significantly expedite diagnosis, streamlining processes, and hastening life-saving interventions [329]. The widespread adoption of tools like RAPID and Viz.ai in thousands of hospitals signifies their effectiveness. Technologies emerging from unified entities, like Viz.ai or RapidAI, have introduced holistic systems that encompass mobile alerts, secure communication avenues, and remote CT/MRI viewing. By merging care coordination with AI/ML’s prowess, these technologies amplify their impact. However, the onus remains on future technological solutions to be encompassed within integrated clinical platforms. Parallelly, AI-driven post-stroke rehabilitation devices are heralding innovative therapeutic methodologies, broadening recovery prospects for patients grappling with severe neurological damage. Comparative assessments of these tools will be pivotal for informed clinical decisions in the future [329].

Delving deeper, recent studies illuminate an intriguing association between gut dysbiosis and the underpinnings and outcomes of stroke. A wealth of clinical and preclinical investigations over the last decade have posited a link between gut imbalance and several stroke risk factors, including obesity, hypertension, and cardiovascular issues, among others [346]. Emerging insights suggest that post-stroke outcomes might be detrimentally affected by gut dysbiosis. Mechanisms precipitating this encompass a gamut of biological processes, including immune-driven systemic inflammation. These cascading effects compound poor outcomes post-stroke, emphasizing the potential of the gut microbiome as a promising therapeutic target.

In sum, the multi-pronged aspects of stroke management underscore the intricacies involved and the necessity for an integrated approach. From timely interventions and technological innovations to deeper insights into the gut–brain connection, the roadmap to effective stroke care demands a holistic view, ever-evolving research, and global collaborative efforts.

### 8.2. Unveiling the Path Forward: Stroke Research, Cellular Mechanisms, and Clinical Advances

The global health landscape recognizes stroke as the second predominant cause of both mortality and long-term disability. Alongside its clinical implications, the economic ramifications of stroke are vast. Over the past quarter century, the realm of stroke research has made substantial strides, encompassing advancements in animal experimental models, therapeutic agents, clinical trials, and rehabilitation strategies. Yet, a significant lacuna in our understanding of effective stroke treatments remains. Notwithstanding our enhanced comprehension of stroke pathophysiology, the chasm between research discoveries and their clinical applications poses a formidable challenge. The bulk of investigations have centered on re-establishing cerebral blood flow and attenuating post-ischemic neuronal deficits. But the essence lies in ensuring consistent, reliable data, mapping underlying therapeutic mechanisms, and enhancing the translational relevance of pre-clinical findings before venturing into the clinical sphere. Recent years have witnessed significant breakthroughs in stroke research. Enhancements in the choice of animal models, imaging modalities, and methodological advancements have unveiled numerous potential drug targets and therapeutic strategies. Nonetheless, the full translational promise of stroke research remains to be thoroughly explored [347].

A focal point of cerebral health is the microglia, sentinel cells intrinsic to preserving brain homeostasis. Their dual roles in both exacerbating and mitigating ischemic stroke damage, contingent upon the stroke’s temporal phase, draw significant attention. Novel insights underscore the microglia’s plastic nature and its adaptability to diverse functional phenotypes within the compromised brain. This adaptability extends from exacerbating inflammatory responses to fostering tissue regeneration. Therapeutic endeavors thus need to delicately balance these dual roles, emphasizing the necessity of comprehending the intricate regulatory networks governing microglial activation. Potential interventions, such as modulating the shift from M2 to M1 phenotypes in the microglia post-acute phase, offer promising therapeutic avenues. Yet, it is paramount to decipher the mechanisms steering microglia polarization within compromised tissues and avoid broad-spectrum immunosuppression that indiscriminately inactivates microglial subtypes. In essence, therapeutics targeting microglia should not only prioritize acute-phase neuroprotection but also accentuate post-acute regenerative strategies [348].

The intricacies of brain maturation serve as a cornerstone for cognitive development, underscored by neuroplasticity’s manifold manifestations. While adult neurogenesis frequently occupies the research limelight, other neural alterations—including neurogenesis sans division and morphological and neurophysiological shifts—offer fertile grounds for therapeutic interventions, enabling sustained cerebral health across an individual’s lifespan. Recent research efforts denote the attainability of such objectives, albeit their cellular substrates exhibit significant complexity. Emphasizing the value of comparative approaches, it is imperative to synergize findings across varied species to enrich our understanding of human neural plasticity. The nuances of comparative neuroscience are thus pivotal in deciphering fundamental biological processes in the human cerebrum [349].

The incorporation of point-of-care testing (POCT) in predicting stroke prognosis is an evolving facet of contemporary clinical practice. Although prevalent POCTs cater to general emergency department patients, bespoke stroke POCT devices are in their nascent stages. As the diagnosis and management of stroke are intrinsically time-sensitive, integrating POCTs can substantially elevate patient care during critical phases. The present-day scientific milieu is poised to craft tailored stroke POCT devices, leveraging cutting-edge immunoassays to discern stroke-specific biomarkers. With the burgeoning landscape of biomarker research, harnessing contemporary analytical instruments is indispensable for integrating POCTs seamlessly into stroke care paradigms [350].

In sum, the path ahead in stroke research, management, and clinical applications mandates continuous exploration, collaboration, and a steadfast commitment to patient-centric care. The intersections of cellular mechanisms, innovative technologies, and clinical practice beckon a holistic approach, fostering advancements in stroke therapy and prognosis Figure 3.

Immediate Response:Upon the onset of stroke symptoms, the emergency medical services (EMS) dispatch is contacted (t = 0).Initial Assessment by EMS:The EMS team initiates a ‘case entry protocol’ to assess the patient’s medical condition based on specific parameters.Patient Transportation:The patient is transported to the nearest hospital via ambulance, a process that ideally takes 15 min or less (t ≤ 15 min).Traditional Stroke Identification Protocol:Best Case: If the EMS has already diagnosed the patient with a stroke, they are directly sent to a specialized stroke unit.Alternate Case: If not pre-diagnosed by EMS, the patient is directed to the Emergency Department (ED), where they await assessment alongside patients with various medical emergencies (t ≤ 30 min).A neurologist then evaluates the suspected stroke patient (t ≤ 60 min). The most prevalent neurological assessment tool is the NIHSS, followed by a CT/MRI scan for an initial diagnosis (t ≤ 90 min).The classification of the stroke subtype is then determined, with the process taking up to 120 min (t ≤ 120 min). Classification methods include the Trial of Org 10172 in Acute Stroke Treatment (TOAST), National Institute of Neurological Disorders and Stroke (NINDS), or Oxford Community Stroke Project (OCSP) schemes.The administration of the tissue plasminogen activator (tPA) typically occurs within 3 h from the onset of stroke symptoms (t ≤ 180 min).POCT-Enhanced Stroke Identification Protocol:The utilization of pre-hospital POCT devices can expedite the timeline from the onset of stroke symptoms to the examination by a neurologist, reducing the wait time to 20 min or less (t ≤ 20 min).Leveraging in-hospital POCT tools can decrease the time required for a CT/MRI scan (t ≤ 45 min).Most critically, the window for administering tPA can be significantly shortened to between 60 and 90 min post-symptom onset (60 min ≤ t ≤ 90 min).

Advancing our understanding of the cognitive dynamics following a transient ischemic attack (TIA) remains a compelling scientific imperative. Delving deeper into the cognitive trajectories post-TIA is critical, especially in discerning how the determinants of delayed-onset dementia (which emerges months or even years after a TIA) diverge from those influencing early-onset or possibly transient cognitive impairments appearing soon after the incident. A nebulous area also exists around the underlying processes that link vascular risk factors, such as diabetes or atrial fibrillation, with the ensuing risk of dementia. Additionally, the potential malleability of this risk based on intensified control of these risk factors or stroke preventive measures remains a question [351].

For comprehensive insights, rigorous research designs are essential. This involves longitudinal investigations paired with superior brain imaging techniques and cognitive evaluations. Parallelly, meticulous oversight of secondary preventive approaches will be invaluable. A notable challenge in extending these findings lies in their generalizability. For instance, the OXVASC study’s outcomes predominantly focus on white study cohorts, potentially limiting the extrapolation of results to diverse ethnicities and healthcare frameworks. The nuanced task of selecting appropriate “controls” in research for a credible comparison with TIA-affected individuals also beckons attention. Historically, research studies have relied on community members who willingly participated in their experiments as control groups. Yet, this practice begs the question of the representativeness of such volunteers, particularly concerning their vascular and TIA risks or historical health profiles, and the potential predisposition towards cognitive decline [351].

This selection could inadvertently lead to a bias, wherein control groups inherently consist of individuals who might have heightened anxieties about their cognitive health risks. A viable countermeasure to this predicament might involve establishing expansive, longitudinal cohorts. Such cohorts could aim to mirror the broader demographic makeup as observed in studies like *The Canadian Longitudinal Study of Aging* or *The 1946 National Survey of Health and Development cohort from the UK*. Alternatively, some researchers might gravitate towards curating specific groups based on their projected cognitive decline risks. A case in point is the UK’s PREVENT cohort, which is strategically enrolling participants spanning high, medium, and low susceptibility brackets for late-onset Alzheimer’s disease. Striking a harmonious balance remains a challenge: researchers must weigh the merits of a large, encompassing population-based research design against a more targeted population that allows for a deeper dive into individual assessments and phenotypic elucidations [351].

In essence, as we endeavor to elucidate the complexities surrounding TIA and its cognitive ramifications, it is paramount that research methodologies evolve with an eye on inclusivity, precision, and generalizability to truly benefit a global audience.

## Figures and Tables

**Figure 1 biomedicines-11-02617-f001:**
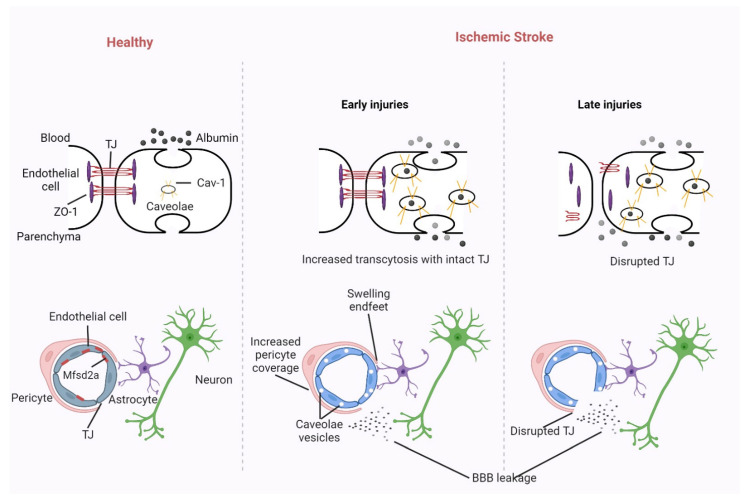
Illustrative Overview of Blood–Brain Barrier Dynamics in the Wake of Acute Ischemic Stroke.

**Figure 2 biomedicines-11-02617-f002:**
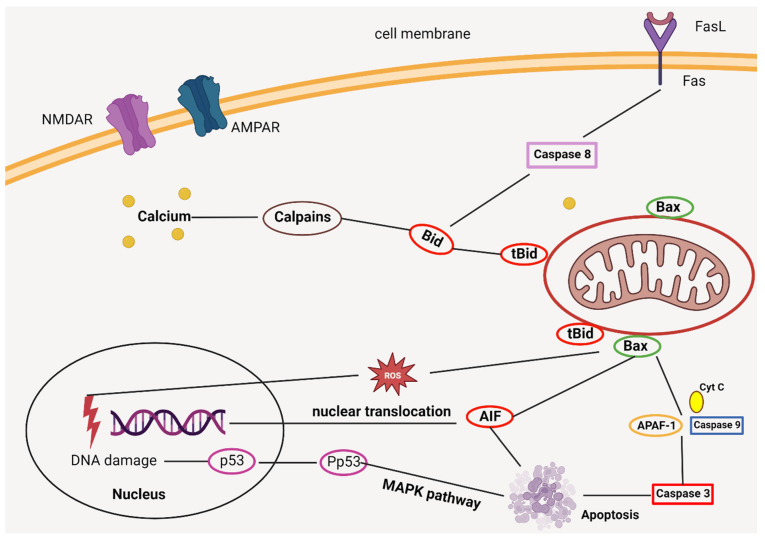
The intricate processes leading to ischemic neuronal cell death involve various molecular pathways and interactions. When NMDA receptors (NMDARs) and AMPA receptors (AMPARs) are stimulated, there is an elevation in the levels of calcium within the cell’s cytoplasm. This heightened calcium concentration triggers the activation of enzymes known as calpains and induces dysfunction within the mitochondria. Simultaneously, the binding of Fas ligands (FasL) to their counterparts, the Fas death receptors, sets off the activation of a protein called caspase 8. These activated calpains, together with caspase 8, collaborate to cleave a protein named Bid, transforming it into its truncated version, tBid. Once formed, tBid associates with another protein, Bax, on the mitochondrial membrane. This interaction is critical as it results in the creation of pores in the membrane, leading to the expulsion of several vital molecules: cytochrome c (Cyt c), apoptosis-inducing factor (AIF), and the harmful reactive oxygen species (ROS). After their release, both ROS and AIF relocate to the cell’s nucleus. Here, they play a pivotal role in damaging the DNA and initiating specific neuronal cell death pathways. A prime example of such a pathway is the phosphorylation of the protein p53, which, when phosphorylated (Pp53), activates the MAPK signaling route, pushing the cell towards apoptosis. Additionally, the expelled cytochrome c from the mitochondria plays a role outside its traditional function. In the cell’s cytoplasm, cytochrome c collaborates with the apoptotic protein activating factor-1 (APAF-1) and procaspase 9 to assemble into a complex known as the apoptosome. This structure is integral to the internal apoptotic pathway, further cementing the neuron’s fate towards programmed cell death.

**Figure 3 biomedicines-11-02617-f003:**
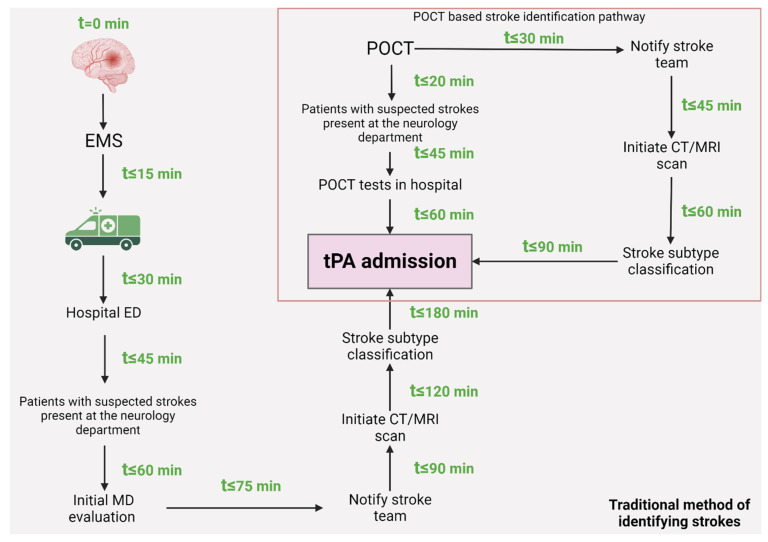
Comparison of Traditional Stroke Prognostic Management vs. Enhanced Approach Using Point-of-Care-Tests (POCTs).

**Table 2 biomedicines-11-02617-t002:** Stages of motor recovery.

Criteria/Stage	Cognitive Stage	Associative Stage	Automatic Stage	Citations
Primary Focus	In this stage of motor learning, the therapist helps the patient learn a piece of work.	Therapist assists the patient in task performance.	Patient is skilled and can perform tasks.	[280]
Decision Making	Decision making is based on “What to do?”	Decision making is based on “How to do a task?”	Decision making is based on “How to succeed?”	[281]
Task Execution	Learner constructs a motor program.	Patient performs and corrects errors; self-evaluation is promoted.	Complex and challenging tasks are performed to gain retention.	[254,280]
Self-evaluation and Feedback	Examine the task’s demands and his ability to complete it.	Continuity proven when error becomes consistent.	Select appropriate feedback.	[280,281]
Task Perception and Memory Recall	Identify the elements and recall the memory.	Emphasize the proprioception “feel of movement”.	Organize practice, self-evaluation and correction, gain retention.	[254,282]
Practice and Problem-solving	The patient then begins practicing the task, identifying and resolving problems.	Assist the learner with self-evaluation and decision-making skills.	Focus on the competitive aspect of the skills.	[280,281,282]

## Data Availability

All data are available on PubMed.

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
