# Peer review of "Integrative Approaches in Acute Ischemic Stroke: From Symptom Recognition to Future Innovations"

_biomedicines, 2023, doi:10.3390/biomedicines11102617_

Round 1

Reviewer 1 Report

Amidst the prevalent cerebrovascular diseases, acute ischemic stroke stands out, carrying significant global health and socio-economic implications. Prompt diagnosis and intervention are paramount, making understanding the varied stroke onset symptoms crucial. With symptoms varying based on the cerebral region affected, a multi-disciplinary team's role becomes instrumental in timely recognition and treatment. Neuroimaging and neuroradiology have evolved dramatically over the years, with various modalities offering distinct advantages. Properly interpreting these images, especially in recognizing cerebral artery occlusions, is vital for effective therapy planning. On the treatment front, the pharmacological approach, particularly fibrinolytic therapy, has its merits and challenges. Endovascular thrombectomy, a game-changer in stroke management, has witnessed significant advances, with technologies like stent retrievers and aspiration catheters playing pivotal roles. For select patients, combining pharmacological and endovascular strategies offers evidence-backed benefits. As the authors aimed for holistic patient management, the emphasis is not just on medical intervention but also on physical therapy, mental health, and community engagement. The future holds promising innovations, with artificial intelligence posed to reshape stroke diagnostics and treatments. Bridging the gap between groundbreaking research and clinical practice remains a challenge, urging continuous collaboration and research.

This review is well written and brings new knowledge to the topic of ischemic stroke. Nevertheless, in this review, it is worth referring to the current publications in the literature, which emphasize that risk factors for ischemic stroke are also risk factors for atrial fibrillation. The same risk factors are also responsible for poor prognosis in patients with acute ischemic stroke (Wańkowicz et al, Boehme et al).

Author Response

Dear Reviewer,

Thank you for your positive feedback and kind suggestions,

We’ve added the correlation between ischemic stroke risk factors and atrial fibrillation risk factors. Moreover, poor prognosis of patients with acute ischemic stroke and the underlying risk factors are now mentioned.

New data was added regarding future perspectives of stroke management and an in-depth analyses of inflammatory mechanisms

Thank you for your significant contribution!

Reviewer 2 Report

overall the work is well written, however I suggest making some changes. First of all, I recommend rewriting the abstract as it is unclear and disorganized. I recommend reducing the part dedicated to the historical chronology of the stroke and giving more emphasis to the present and future. In the neuroimaging section I recommend adding a paragraph on the role of collateral circles, a very current and challenging topic. There are various interesting and recent works on which to draw inspiration.
I found the part dedicated to rehabilitation very interesting. Finally, I recommend expanding the paragraph dedicated to neuroinflammation.  

Icona di Verificata con community

Author Response

Dear Reviewer,

Thank you for your positive feedback and kind suggestions,

We’ve rewritten the abstract so it can be more clear, historical chronology part is now reduced and future perspectives are discussed in various parts of the manuscript.

A paragraph regarding the role of collateral circles is now mentioned and we’ve added an in-depth analyses of neuroinflammatory mechanisms  

Thank you for your significant contribution!